# Difficult Examples Hurt Unsupervised Contrastive Learning: A Theoretical Perspective

**Yi-Ge Zhang**[1*]    **Jingyi Cui**[1*]    **Qiran Li**[2†]    **Yisen Wang**[1,3‡]

[1]State Key Lab of General AI, School of Intelligence Science and Technology, Peking University
[2]School of Engineering, Hong Kong University of Science and Technology
[3]Institute for Artificial Intelligence, Peking University

## Abstract

Unsupervised contrastive learning has shown significant performance improvements in recent years, often approaching or even rivaling supervised learning in various tasks. However, its learning mechanism is fundamentally different from supervised learning. Previous works have shown that difficult examples (well-recognized in supervised learning as examples around the decision boundary), which are essential in supervised learning, contribute minimally in unsupervised settings. In this paper, perhaps surprisingly, we find that the direct removal of difficult examples, although reduces the sample size, can boost the downstream classification performance of contrastive learning. To uncover the reasons behind this, we develop a theoretical framework modeling the similarity between different pairs of samples. Guided by this framework, we conduct a thorough theoretical analysis revealing that the presence of difficult examples negatively affects the generalization of contrastive learning. Furthermore, we demonstrate that the removal of these examples, and techniques such as margin tuning and temperature scaling can enhance its generalization bounds, thereby improving performance. Empirically, we propose a simple and efficient mechanism for selecting difficult examples and validate the effectiveness of the aforementioned methods, which substantiates the reliability of our proposed theoretical framework.

## 1 Introduction

Contrastive learning has demonstrated exceptional empirical performance in the realm of unsupervised representation learning, effectively learning high-quality representations of high-dimensional data using substantial volumes of unlabeled data by aligning an anchor point with its augmented views in the embedding space (Caron et al., 2020; Chen et al., 2020b;c; 2021; He et al., 2020). Unsupervised contrastive learning may own quite different working mechanisms from supervised learning, as discussed in Joshi & Mirzasoleiman (2023). For example, difficult examples (also known as difficult-to-learn examples in Joshi & Mirzasoleiman (2023)), which contribute the most to supervised learning, contribute the least or even negatively to contrastive learning performance. They

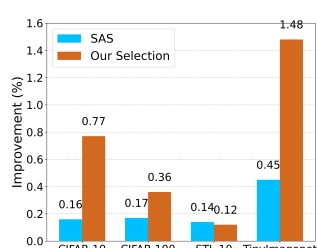

Figure 1: Excluding difficult examples improves unsupervised contrastive learning.

show that on image datasets such as CIFAR-100 and STL-10, excluding 20%-40% of the examples does not negatively impact downstream task performance. More surprisingly, their results showed, but somehow failed to notice, that excluding these samples on certain datasets like STL-10 can lead to performance improvements in downstream tasks.

Taking a step further beyond their study, we find that this surprising result is not just a specialty of a certain dataset, but a universal phenomenon across multiple datasets. Specifically, we run SimCLR on the original CIFAR-10, CIFAR-100, STL-10, and TinyImagenet datasets, the SAS core subsets

---
[*]Equal contribution
[†]Work was done when he was an undergraduate intern at Peking University
[‡]Corresponding author: Yisen Wang (yisen.wang@pku.edu.cn)

(Joshi & Mirzasoleiman, 2023) selected with a deliberately tuned size, and a subset selected by a sample removal mechanism to be proposed in this paper. In Figure 1, we report the gains of linear probing accuracy by using the subsets compared with the original datasets. We see that on all these benchmark datasets, excluding a certain fraction of examples results in comparable and even better downstream performance. This result is somewhat anti-intuitive because deep learning models trained with more samples, benefiting from lower sample error, usually perform better. Yet our observation indicates that difficult examples can hurt unsupervised contrastive learning performances. This observation naturally raises a question:

> *What is the mechanism behind difficult examples impacting the learning process of unsupervised contrastive learning?*

To comprehensively characterize such impact, we first develop a theoretical framework, i.e., the similarity graph, to describe the similarity between different sample pairs. Specifically, pairs containing difficult samples, termed as difficult pairs, exhibit higher similarities than other different-class pairs. Based on this similarity graph, we derive the linear probing error bounds of contrastive learning models trained with and without difficult samples, proving that the presence of difficult examples negatively affects performance. Next, we prove that the most straightforward idea of directly removing difficult examples improves the generalization bounds. Further, we also theoretically demonstrate that commonly used techniques such as margin tuning (Zhou et al., 2024) and temperature scaling (Khaertdinov et al., 2022; Kukleva et al., 2023; Zhang et al., 2021) mitigate the negative effects of difficult examples by modifying the similarity between sample pairs from different perspectives, thereby improving the generalization bounds. Experimentally, we propose a simple but effective mechanism for selecting difficult samples that does not rely on pre-trained models. The performance improvements achieved by addressing difficult samples through the aforementioned methods align with our theoretical analysis of the generalization bounds.

The contributions of this paper are summarized as follows:

- We find that removing certain training examples boosts the performance of unsupervised contrastive learning is a universal empirical phenomenon on multiple benchmark datasets. Through a mixing-image experiment, we conjecture that the removal of difficult examples is the cause.

- We design a theoretical framework that models the similarity between different pairs of samples to characterize how difficult samples in contrastive learning affect the generalization of downstream tasks. Based on this framework, we theoretically prove that the existence of difficult samples hurts contrastive learning performances.

- We theoretically analyze how possible solutions, i.e. directly removing difficult samples, margin tuning, and temperature scaling, can address the issue of difficult examples by improving the generalization bounds in different ways.

- In experiments, we propose a simple and efficient mechanism for selecting difficult examples and validate the effectiveness of the aforementioned methods, which substantiates the reliability of our proposed theoretical framework.

## 2 DIFFICULT EXAMPLES HURT: A MIXING IMAGE EXPERIMENT

We start this section by revealing that difficult examples do hurt contrastive learning performances through a proof-of-concept toy experiment.

The concept of difficult examples is primarily motivated by supervised learning, where the learning difficulty depends on factors such as data quality (Li et al., 2019; Shin et al., 2020), sample neighbors (Chen et al., 2020a), and category distribution (Cui et al., 2019; Santiago et al., 2021). Difficult samples (also termed as difficult-to-learn examples or hard samples) usually lie around the Bayes decision boundary and therefore render high loss or large gradient norm during training (Joshi & Mirzasoleiman, 2023), e.g., a cat that looks like a dog or a blurry image. In the purely unsupervised learning paradigm, as labels and decision boundaries are absent, we use sample similarity to represent difficult samples in a pair-wise manner, i.e., difficult sample pairs are inter-class sample pairs with high similarity, which leads to their susceptibility to being wrongly clustered during

self-supervised pre-training. It is somewhat related to hard negative samples, a pure unsupervised learning concept defined as highly similar negative samples to the anchor point, but is different in nature. (See Appendix A.1 for more discussions.)

In real datasets, as difficult examples rely on the specific classifier trained in the supervised learning manner, we can not precisely know the ground truth difficult examples. Therefore, we in turn add additional difficult examples and observe the effects of these examples. Specifically, we generate a mixing-image dataset containing more difficult samples by mixing a $\omega$ fraction of images on CIFAR-10 dataset at the pixel level (these samples lie around the class difficult), termed as $\omega$-Mixed CIFAR-10 datasets. Then, we train the representative contrastive learning algorithm SimCLR (Chen et al., 2020b) on the original, 10%-, and 20%-Mixed CIFAR-10 datasets using ResNet18 model. We report the linear probing accuracy in Figure 2.

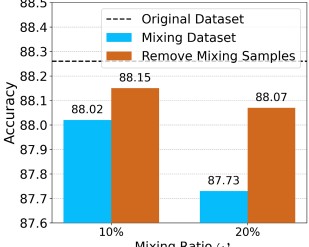

Figure 2: Excluding (mixed) difficult examples improves performance.

Compared with the model trained on the original dataset, we find that with the mixed difficult examples included in the training dataset, the performance of contrastive learning drops. This result indicates that the (mixed) difficult samples significantly negatively impact contrastive learning. As the mixing ratio $\omega$ increases, the performance drops, indicating that more difficult examples lead to worse contrastive learning performances.

Moreover, we show that removing the (mixed) difficult samples can boost performance. Specifically, we compare performance on the Mixed CIFAR-10 datasets with that on the datasets removing the mixed examples. As shown in Figure 2, despite being trained with a smaller sample size, models trained on datasets removing the mixed examples perform better than the ones trained with the mixed examples, which further verifies that difficult examples hurt unsupervised contrastive learning, and removal of these difficult examples can boost learning performance.

## 3 THEORETICAL CHARACTERIZATION OF WHY DIFFICULT EXAMPLES HURT CONTRASTIVE LEARNING

In this section, to explain why difficult examples negatively impact the performance of contrastive learning, we provide theoretical evidence on generalization bounds. In Section 3.1 we present the necessary preliminaries that lay the foundation for our theoretical analysis. In Section 3.2, we introduce the similarity graph describing difficult examples. In Section 3.3, we respectively derive error bounds of contrastive learning with and without difficult examples.

### 3.1 PRELIMINARIES

**Notations.** Given a natural data $\bar{x} \in \bar{\mathcal{X}} := \mathbb{R}^d$, we denote the distribution of its augmentations by $\mathcal{A}(\cdot|\bar{x})$ and the set of all augmented data by $\mathcal{X}$, which is assumed to be finite but exponentially large. For mathematical simplicity, we assume class-balanced data with $n$ denoting the number of augmented samples per class and $r + 1$ denoting the number of classes, hence $|\mathcal{X}| = n(r + 1)$. Let $n_d$ represent the number of difficult examples per class and $\mathbb{D}_d$ the set of difficult examples. In addition, we denote $k$ as the feature dimension in contrastive representation learning.

**Similarity Graph (Augmentation Graph).** As described in HaoChen et al. (2021), an augmentation graph $\mathcal{G}$ represents the distribution of augmented samples, where the edge weight $w_{xx'}$ signifies the joint probability of generating augmented views $x$ and $x'$ from the same natural data, i.e., $w_{xx'} := \mathbb{E}_{\bar{x}\sim\bar{\mathcal{P}}}[\mathcal{A}(x|\bar{x})\mathcal{A}(x'|\bar{x})]$, where $\bar{\mathcal{P}}$ denotes the distribution of natural data. The total probability across all pairs of augmented data sums up to 1, i.e., $\sum_{x,x'\in\mathcal{X}} w_{xx'} = 1$. The adjacency matrix of the augmentation graph is denoted as $\boldsymbol{A} = (w_{xx'})_{x,x'\in\mathcal{X}}$, and the normalized adjacency matrix is $\bar{\boldsymbol{A}} = \boldsymbol{D}^{-1/2}\boldsymbol{A}\boldsymbol{D}^{-1/2}$, where $D := \text{diag}(w_x)_{x\in\mathcal{X}}$, and $w_x := \sum_{x'\in\mathcal{X}} w_{xx'}$. The concept of augmentation graph is further extended to describe similarities beyond image augmentation, such as cross-domain images (Shen et al., 2022), multi-modal data (Zhang et al., 2023), and labeled examples (Cui et al., 2023).

**Contrastive losses.** For theoretical analysis, we consider the spectral contrastive loss $\mathcal{L}(f)$ proposed by HaoChen et al. (2021) as a good performance proxy for the widely used InfoNCE loss

$$\mathcal{L}_{\mathrm{Spec}}(f) := -2 \cdot \mathbb{E}_{x,x^+}[f(x)^\top f(x^+)] + \mathbb{E}_{x,x'}\left[\left(f(x)^\top f(x')\right)^2\right], \tag{1}$$

where $x$, $x^+$, and $x'$ represent the anchor, positive sample, and negative sample, respectively. As proved in Balestriero & LeCun (2022); Johnson et al. (2022); Tan et al. (2024), the spectral contrastive loss and the InfoNCE loss share the same population minimum with variant kernel derivations. Further, the spectral contrastive loss is theoretically shown to be equivalent to the matrix factorization loss. For $F = (u_x)_{x \in \mathcal{X}}$, where $u_x = w_x^{1/2} f(x)$, the matrix factorization loss is:

$$\mathcal{L}_{\mathrm{mf}}(F) := \|\bar{A} - FF^\top\|_F^2 = \mathcal{L}_{\mathrm{Spec}}(f) + const. \tag{2}$$

## 3.2 MODELING OF DIFFICULT EXAMPLES

We start by introducing a similarity graph, to describe the relationships between various samples. In contrastive learning, examples are used in a pairwise manner, so we define difficult sample pairs as sample pairs that include at least one difficult sample. As difficult examples lie around the decision boundary, they should have higher augmentation similarity to examples from different classes. Therefore, it is natural for us to define the difficult pairs as different-class sample pairs with higher similarity. Correspondingly, easy pairs are defined as different-class sample pairs containing no difficult samples, or different-class sample pairs with lower similarity.

Specifically, we define the augmentation similarity between a sample and itself as $1$. Then we assume the similarity between same-class samples is $\alpha$ (Figure 3(a)), the similarity between a sample (conceptually far away from the class boundary) and all samples from other classes is $\beta$ (Figure 3(b)), and the similarity between different-class boundary samples (conceptually close to the class boundary) is $\gamma$ (Figure 3(c)). Naturally, we have $\beta < \gamma < \alpha < 1$.

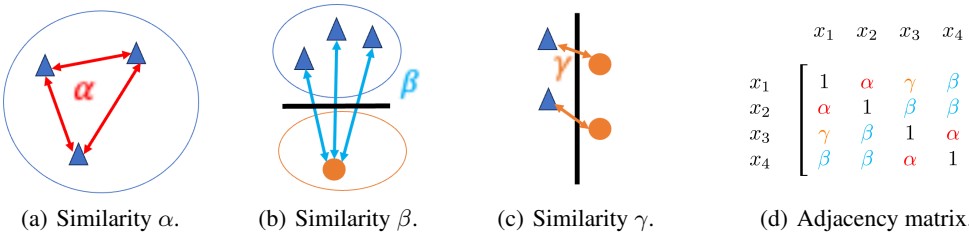

(a) Similarity $\alpha$.    (b) Similarity $\beta$.    (c) Similarity $\gamma$.    (d) Adjacency matrix.

Figure 3: Modeling of difficult examples. The similarity between same-class samples is $\alpha$ (a), the similarity between different-class difficult samples is $\gamma$ (c), and the similarity between other samples is $\beta$ (b). The adjacency matrix of a 4-sample subset is shown in (d).

In Figure 3(d), we illustrate our modeling of adjacency matrix through a 4-sample subset $\mathbb{D}_4 := x_1, x_2, x_3, x_4$, where $x_1$ and $x_2$ belong to Class 0, and $x_3$ and $x_4$ belong to Class 1. We define $x_1$ and $x_3$ as difficult samples (assuming these two samples are distributed around the classification boundary as depicted in Figure 3(c)), i.e. $x_1, x_3 \in \mathbb{D}_d$. Conversely, we define $x_2$ and $x_4$ (assuming these samples are distributed far from the classification boundary) as easy samples, i.e. $x_2, x_4 \in \mathbb{D}_4 \setminus \mathbb{D}_d$. The relationship between each pair of samples in $\mathbb{D}_4$ can be mathematically formulated as an adjacency matrix shown in Figure 3(d).

In addition, the above modeling could be relaxed by adding random terms to the similarity values. Specifically, for some constant $\epsilon > 0$, for a similarity matrix $A = (\tilde{a}_{ij})$, we replace $a_{ij}$ with $\tilde{a}_{ij} = a_{ij} + \epsilon \cdot \varepsilon_{ij}$ for $i \neq j$, where $a_{ij}$ takes values in $\{\alpha, \beta, \gamma\}$, $\varepsilon_{ij} = \varepsilon_{ji}$ are i.i.d. random variables with mean 0 and variance 1. We discuss the relaxation in detail in Section B.3.

In what follows, our theoretical analysis is based on the generalized similarity graph containing $|\mathcal{X}| = n(r+1)$ samples. The formal definition of the generalized adjacency matrix is in Appendix B.

### 3.3 Error Bounds with and without Difficult Examples

Based on the similarity graph in Section 3.2, we derive the linear probing error bounds for contrastive learning models trained with and without difficult examples in Theorems 3.3 and 3.4. We mention that we adopt the label recoverability (with labeling error $\delta$) and realizability assumptions from HaoChen et al. (2021).

**Assumption 3.1** (Labels are recoverable from augmentations). *Let $\bar{x} \sim \mathcal{P}_{\bar{\mathcal{X}}}$ and $y(\bar{x})$ be its label. Let the augmentation $x \sim \mathcal{A}(\cdot|\bar{x})$. We assume that there exists a classifier $g$ that can predict $y(\bar{x})$ given $x$ with error at most $\delta$, i.e. $g(x) = y(\bar{x})$ with probability at least $1 - \delta$.*

**Assumption 3.2** (Realizability). *Let $\mathcal{F}$ be a hypothesis class containing functions from $\mathcal{X}$ to $\mathbb{R}^k$. We assume that at least one of the global minima of $\mathcal{L}_{\mathrm{Spec}}$ belongs to $\mathcal{F}$.*

Assumption 3.1 indicates that labels are recoverable from the augmentations, and Assumption 3.2 indicates that the universal minimizer of the population spectral contrastive loss can be realized by the hypothesis class. The proofs are shown in Appendix B.1.

**Theorem 3.3** (Error Bound without Difficult Examples). *Denote $\mathcal{E}_{\mathrm{w.o.}}$ as the linear probing error of a contrastive learning model trained on a dataset without difficult examples. Then*

$$\mathcal{E}_{\mathrm{w.o.}} \leq \frac{4\delta}{1 - \frac{1-\alpha}{(1-\alpha)+n\alpha+nr\beta}} + 8\delta. \tag{3}$$

**Theorem 3.4** (Error Bound with Difficult Examples). *Denote $\mathcal{E}_{\mathrm{w.d.}}$ as the linear probing error of a contrastive learning model trained on a dataset with $n_d$ difficult examples per class. Then if $r + 1 \leq k < n_d + r + 1$, there holds*

$$\mathcal{E}_{\mathrm{w.d.}} \leq \frac{4\delta}{1 - \frac{(1-\alpha)+r(\gamma-\beta)}{(1-\alpha)+n\alpha+nr\beta+r(\gamma-\beta)}} + 8\delta. \tag{4}$$

**Discussions.** By comparing Theorems 3.3 and 3.4, also considering that $\frac{(1-\alpha)+r(\gamma-\beta)}{(1-\alpha)+n\alpha+nr\beta+r(\gamma-\beta)} > \frac{1-\alpha}{(1-\alpha)+n\alpha+nr\beta}$, we see the presence of difficult examples leads to a strictly worse linear probing error bound for a contrastive learning model. Moreover, more challenging difficult examples (larger $\gamma - \beta$) result in worse error bounds, and more difficult examples (larger $n_d$) induce a larger range of embedding dimension $k$. Specifically, when $\gamma = \beta$, i.e., no difficult examples exist, the bound in Theorem 3.4 reduces to that in Theorem 3.3.

Intuitively, through the augmentation graph, contrastive learning could be understood as a spectral clustering problem (HaoChen et al., 2021). As the difficult examples lie very close to the classification boundary, they could fall into the wrong clusters during self-supervised pre-training. In the downstream applications, the wrongly clustered examples provide false prior knowledge to the downstream classification, which harms the performance of all test samples.

## 4 Theoretical Characterization on Eliminating Effects of Difficult Examples

Building on the above unified theoretical framework, we theoretically analyze that directly removing difficult samples (Section 4.1), margin tuning (Section 4.2), and temperature scaling (Section 4.3) can handle difficult examples by improving the generalization bounds in different ways.

### 4.1 Removing Difficult Samples

In Figures 1 and 2, empirical experiments demonstrated that removing difficult samples can improve learning performance. Corollary 4.1 provides a theoretical explanation for this counter-intuitive phenomenon based on our established framework.

**Corollary 4.1.** *Denote $\mathcal{E}_{\mathrm{R}}$ as the linear probing error of a contrastive learning model trained on a selected subset removing all difficult examples $\mathbb{D}_d$. Then there holds*

$$\mathcal{E}_{\mathrm{R}} \leq \frac{4\delta}{1 - \frac{1-\alpha}{(1-\alpha)+(n-n_d)\alpha+(n-n_d)r\beta}} + 8\delta. \tag{5}$$

Corollary 4.1 shows that when the difficult examples are removed, the linear probing error bound has the same form as the case where no difficult examples are present (Theorem 3.3), but with $n$ replaced by $n - n_d$. Compared with the case without removing difficult examples (Theorem 3.4), the bound in equation 5 is smaller than that in equation 4 when $\gamma - \beta > \frac{n_d(1-\alpha)}{r(n-n_d)}$. This indicates that removing difficult examples enhances the error bound when these samples are significantly harder than the easy ones (i.e., large $\gamma - \beta$) or when the number of difficult samples is small (i.e., small $n_d$).

## 4.2 MARGIN TUNING

Aside from sample removal, we also consider using the margin tuning technique to deal with difficult examples. Specifically, we add additional margin parameters to the similarity of difficult pairs in the loss function (see Eq. 14). Here, we delve into how margin tuning can enhance the generalization in the presence of difficult examples.

**Theorem 4.2.** *The margin tuning loss is equivalent to the matrix factorization loss*

$$\mathcal{L}_{\mathrm{mf-M}}(F) := \|(\bar{A} - \bar{M}) - FF^\top\|_F^2, \tag{6}$$

*where $\bar{A}$ is the normalized adjacency matrix, and $\bar{M}$ is the normalized margin matrix.*

Theorem 4.2 indicates that adjusting margins alters the similarity graph by subtracting a normalized margin matrix $\bar{M}$ from the normalized similarity matrix $\bar{A}$. Intuitively, by subtracting the additional similarity values of difficult examples with appropriately chosen margins, the remaining values will match those of easy examples. Specifically, in the following Theorem 4.3, we show that properly chosen margins can eliminate the negative impact of difficult examples.

**Theorem 4.3.** *Denote $\mathcal{E}_{\mathrm{M}}$ as the linear probing error for the margin tuning loss equation 31 trained on a dataset with difficult samples $\mathbb{D}_d$. If we let*

$$m_{x,x'} = c_0/(c_1^2 c_2) \cdot (\gamma - \beta) \tag{7}$$

*for $y(x) \neq y(x'), x, x' \in \mathbb{D}_d$, where $c_0 := (1-\alpha) + n\alpha + (n-n_d)r\beta$, $c_1 := (1-\alpha) + n\alpha + nr\beta + r(\gamma - \beta)$ and $c_2 := (1-\alpha) + n\alpha + nr\beta$, and $m_{x,x'} = 0$ for $x, x' \notin \mathbb{D}_d$, then we have*

$$\mathcal{E}_{\mathrm{M}} = \mathcal{E}_{\mathrm{w.o.}}. \tag{8}$$

Note that when $n$ is large enough, $m_{x,x'}$ for $x$ or $x' \notin \mathbb{D}_d$ are higher-order infinitesimals relative to equation 7, and primarily affect normalization rather than the core problem. Thus, we focus on cases where $x, x' \in \mathbb{D}_d$ and defer specific forms of other $m_{x,x'}$ values to the proofs for brevity.

Theorem 4.3 shows that with appropriately chosen margins, the linear probing error bound for the margin tuning loss in the presence of difficult examples becomes equivalent to the standard contrastive loss without such examples, as indicated in Theorem 3.3. Since $equation\ 7 > 0$, this suggests applying a positive margin to the difficult example pairs. Additionally, the more challenging the example pairs are (i.e., the larger $\gamma - \beta$), the greater the margin value should be.

## 4.3 TEMPERATURE SCALING

We also consider the widely used temperature scaling technique in eliminating the negative effects of difficult examples. Specifically, we add an additional temperature scaling parameter to the base temperature of difficult pairs in the loss function and assign the base temperature to all the other pairs (see Eq. 15). Here, we investigate how temperature scaling can enhance generalization.

**Theorem 4.4.** *The temperature scaling loss is equivalent to the matrix factorization loss*

$$\mathcal{L}_{\mathrm{mf-T}}(F) := \|\boldsymbol{T} \odot \bar{A} - FF^\top\|_{wF}^2, \tag{9}$$

*where $\bar{A}$ is the normalized adjacency matrix of similarity graph, $\boldsymbol{T} \odot \bar{A}$ is the element-wise product of matrices $\boldsymbol{T}$ and $\bar{A}$, and $\|\cdot\|_{wF}$ is the weighted Frobenius norm (specified in the proof).*

Theorem 4.4 shows that adjusting temperatures modifies the similarity graph by multiplying the temperature values with the normalized similarity matrix $\bar{A}$. Intuitively, by scaling the similarity values between difficult examples, we can match these values to those of easy examples, thereby mitigating the negative effects of difficult examples. Specifically, the following Theorem 4.5 outlines the appropriate temperature values to be chosen.

**Theorem 4.5.** *Denote $\mathcal{E}_{\mathrm{T}}$ as the linear probing error for the temperature scaling loss equation 40 trained on a dataset with difficult samples $\mathbb{D}_d$. If we let*

$$\tau_{x,x'} = (c_1/c_2)(\beta/\gamma) \tag{10}$$

*for $y(x) \neq y(x'), x, x' \in \mathbb{D}_d$, where $c_1 := (1-\alpha) + n\alpha + nr\beta + r(\gamma - \beta)$ and $c_2 := (1-\alpha) + n\alpha + nr\beta$, and $\tau_{x,x'} = 1$ for $x, x' \notin \mathbb{D}_d$, then we have*

$$\mathcal{E}_{\mathrm{T}} \leq \frac{4[1 - (n_d/n)^2 + (\gamma/\beta)^2 (n_d/n)^2]\delta}{1 - \frac{1-\alpha}{(1-\alpha)+n\alpha+nr\beta}} + 8\delta. \tag{11}$$

Likewise, here we only focus on the temperature values between difficult examples, and defer the specific forms of other $\tau_{x,x'}$ values to the proofs for brevity.

Theorem 4.5 shows the linear probing error bound of the temperature scaling loss when trained on data containing difficult examples. Specifically, with large $n$ and $n_d/n \to 0$, we have $\mathcal{E}_{\mathrm{T}}/\mathcal{E}_{\mathrm{w.o.}} - 1 \approx O((n_d/n)^2)$ and $\mathcal{E}_{\mathrm{w.d.}}/\mathcal{E}_{\mathrm{w.o.}} - 1 \approx O(1/n)$. This indicates that, when $O(n_d) \lesssim O(n^{1/2})$, $\mathcal{E}_{\mathrm{T}}/\mathcal{E}_{\mathrm{w.o.}} \lesssim \mathcal{E}_{\mathrm{w.d.}}/\mathcal{E}_{\mathrm{w.o.}}$, meaning $\mathcal{E}_{\mathrm{T}}$ converges faster to $\mathcal{E}_{\mathrm{w.o.}}$. Detailed calculations show that when $n_d < \sqrt{\frac{r}{(\alpha+r\beta)(\gamma+\beta)}}\beta \cdot n^{1/2}$, there holds $\mathcal{E}_{\mathrm{T}} < \mathcal{E}_{\mathrm{w.d.}}$, which means that temperature scaling improves the error bound. Note that we have approximately $\tau_{x,x'} \propto \beta/\gamma$. This inspires us to choose smaller temperature values for the difficult example pairs. The more difficult the example pairs (smaller $\beta/\gamma$), the smaller the temperature values that should be chosen.

## 5 VERIFICATION EXPERIMENTS

This paper primarily focuses on theoretical analysis, explaining how different samples in contrastive learning impact generalization. The experiments in this part are mainly designed to validate the theoretical insights and demonstrate that the proposed directions for improving performance are sound. The experiments are not intended to achieve state-of-the-art results but rather to confirm the correctness of our theoretical findings. We hope that readers will appreciate the theoretical contributions of this work and not focus excessively on the experimental results.

In Section 5.1, we present an efficient mechanism for selecting difficult samples. We then evaluate the removal of difficult samples (Section 5.2), margin tuning (Section 5.3), and temperature scaling (Section 5.4), all of which are theoretically established to mitigate the impact of these difficult examples. In Section 5.5, we propose a combined method, and discuss the scalability under different paradigms and the connection between difficult samples and long-tail distribution. The specific loss forms can be found in Appendix A.2.

### 5.1 DIFFICULT EXAMPLES SELECTION

In this section, we design a simple yet efficient selection mechanism to validate our theoretical analysis, without relying on additional pretrained models or incurring extra computational overhead (Joshi & Mirzasoleiman, 2023).

To identify difficult sample pairs which from different classes but with high similarity, we compute the cosine similarity of each sample to other samples in the same batch using features before projector mapping. We define $posHigh$ and $posLow$ as percentiles of the similarity sorted in descending order, where $Sim_{posHigh}$ and $Sim_{posLow}$ are the corresponding similarities. Generally, following the characterization in Section 3.2 and Appendix B, we can roughly assume $posHigh$ corresponds to $1/(r+1)$, where $r+1$ is the class number[1]. Sample pairs with cosine similarities above $Sim_{posHigh}$ are considered from the same class. Sample pairs with the similarity between $Sim_{posHigh}$ and $Sim_{posLow}$ are considered as difficult examples. Sample pairs with cosine similarities below $Sim_{posLow}$ are considered as easy-to-learn samples from different classes. Here for $posLow$, we note that when optimizing $\gamma$ of difficult examples, if some easy-to-learn samples are involved, the process will also optimize $\beta$, which is a good thing for the representation learning to

---

[1]We do not need to know the exact label of each class. A rough class number is enough, which can be easily known by clustering.

push samples from different classes further apart. Therefore, we can easily find a value close to the bottom of the sorted similarity for $posLow$, even 100%. Experiments in Figure 4(a) and Figure 4(b) show that our method is not sensitive to the exact values of $posHigh$ and $posLow$.

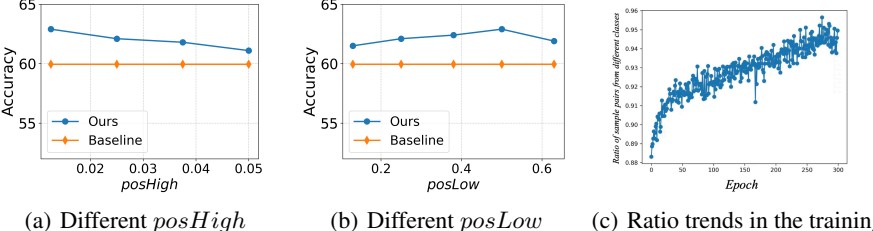

(a) Different $posHigh$       (b) Different $posLow$       (c) Ratio trends in the training

Figure 4: Parameter sensitivity of difficult example interval ends $posHigh$ (4(a)) and $posLow$ (4(b)). Parameter analysis on CIFAR-100: the trend of the ratio of sample pairs from different classes in $(Sim_{posLow}, Sim_{posHigh})$ during the training process (4(c)).

Using this selection mechanism, for an augmented sample pair $(x_i, x_j)$ in the current batch, we define the selecting indicator of difficult pairs as

$$p_{i,j} := \mathbf{1}_{[Sim_{posLow} \le s_{ij} < Sim_{posHigh}]}, \tag{12}$$

where $s_{i,j}$ denotes the cosine similarity between the representations of $x_i$ and $x_j$, and $\mathbf{1}_{[\text{condition}]}$ denotes the indicator function returning 1 if the condition holds and 0 otherwise. For each sample $x_i$, we get a vector $P_i = (p_{i,j})_{j=1}^{2N}$ representing the indicator of difficult pairs. After calculating these indicators for all samples in the current batch, we stack the vectors $P_i$ row-wise to create the selection matrix $\boldsymbol{P}$. In practice, $P_i$ can be computed in parallel, making the computation of $\boldsymbol{P}$ efficient. The elements of $\boldsymbol{P}$ are either 0 or 1, indicating whether pairs are difficult pairs or not.

We can use the class information to verify the proportions of sample pairs from different classes in $(Sim_{posLow}, Sim_{posHigh})$ on CIFAR-10, which can demonstrate the effectiveness of our selection mechanism. As shown in Figure 4(c), along with the progress of training, the ratio of sample pairs from different classes approaches close to 100% within the range $(Sim_{posLow}, Sim_{posHigh})$.

## 5.2 REMOVING DIFFICULT SAMPLES

We here introduce a simple and practical method for removing difficult samples based on our proposed selection mechanism. Eliminating the impact of difficult samples means preventing sample pairs that include difficult samples from interfering with the training process. To achieve this, we use the selection matrix $\boldsymbol{P}$ to identify and remove difficult samples.

Table 1: Classification accuracy with or without removing difficult examples on CIFAR-10, CIFAR-100, STL-10 and TinyImagenet dataset using SimCLR. Results are averaged over three runs.

| Method | CIFAR-10 | CIFAR-100 | STL-10 | TinyImagenet |
|---|---|---|---|---|
| SimCLR (Baseline) | 88.26 | 59.95 | 75.98 | 69.58 |
| SimCLR (Removing) | **89.03** | **60.31** | **76.10** | **71.06** |

It can be observed from Table 1 that removing difficult examples yields a 0.8% performance boost on CIFAR-10, a 0.6% performance boost on CIFAR-100, and a 3.7% performance boost on TinyImagenet compared to the baseline method. We reach the same conclusion as in Joshi & Mirzasoleiman (2023): By removing difficult samples, we can achieve comparable results or even slight improvements over the baseline. However, removing difficult samples may not be the most effective method for handling difficult samples, because it shrinks sample size. Next, we investigate two techniques that handle difficult samples better, margin tuning in Section 5.3 and temperature scaling in Section 5.4.

## 5.3 Margin Tuning on Difficult Samples

To effectively apply margin tuning in line with our theoretical analysis, we adopt a margin tuning factor $\sigma > 0$. For the selected difficult sample pairs identified by the selection matrix $\boldsymbol{P}$, we add a margin $\sigma$ to the similarity values, and for the unselected pairs, we use the original InfoNCE.

Table 2: Classification accuracy with or without margin tuning on CIFAR-10, CIFAR-100, STL-10 and TinyImagenet dataset. Results are averaged over three runs.

| Method | CIFAR-10 | CIFAR-100 | STL-10 | TinyImagenet |
|---|---|---|---|---|
| Baseline | 88.26 | 59.95 | 75.98 | 69.58 |
| MT (All Samples) | 88.52 | 60.09 | 76.02 | 70.06 |
| MT (Selected Samples) | **89.16** | **61.28** | **76.83** | **79.14** |

It can be observed from Table 2 that applying margin tuning to all samples directly only achieves comparable results as the baseline SimCLR, highlighting the importance of the selection mechanism for difficult examples. While applying margin tuning to the selected samples brings consistent performance gains on CIFAR10, CIFAR100, and TinyImageNet. These results validate both the effectiveness of our selection mechanism and the reliability of our analysis on margin tuning.

## 5.4 Temperature Scaling on Difficult Samples

We define the temperature scaling factor $\rho > 0$. Given the base temperature $\tau > 0$, we attach temperature $\rho\tau$ to the selected difficult sample pairs identified by the selection matrix $\boldsymbol{P}$, whereas attach base temperature $\tau$ to the unselected pairs.

Table 3: Classification accuracy with or without temperature scaling on CIFAR-10, CIFAR-100, STL-10 and TinyImagenet dataset. Results are averaged over three runs.

| Method | CIFAR-10 | CIFAR-100 | STL-10 | TinyImagenet |
|---|---|---|---|---|
| Baseline | 88.26 | 59.95 | 75.98 | 69.58 |
| TS (All Samples) | 88.38 | 59.20 | 75.76 | 69.36 |
| TS (Selected Samples) | **89.24** | **61.67** | **76.62** | **78.52** |

It can be observed from Table 3 that applying temperature scaling to all samples directly can even hurt the performance of contrastive learning compared to baseline SimCLR, highlighting the importance of selecting difficult examples. In contrast, applying temperature scaling to the selected samples brings consistent performance gains on CIFAR10, CIFAR100, and TinyImageNet. These results validate both the effectiveness of our selection mechanism and the reliability of our analysis on temperature scaling.

## 5.5 Extensions

**Combined method.** From Sections 4.2 and 4.3, we observe that margin tuning and temperature scaling eliminate the effects of difficult examples in different ways. Therefore, it is natural to combine the two methods, and see if the combined method could reach better performances.

Table 4: Classification accuracy with or without combined method on CIFAR-10, CIFAR-100, STL-10 and TinyImagenet dataset. Results are averaged over three runs.

| Method | CIFAR-10 | CIFAR-100 | STL-10 | TinyImagenet |
|---|---|---|---|---|
| Baseline | 88.26 | 59.95 | 75.98 | 69.58 |
| Margin Tuning | 89.16 | 61.28 | 76.83 | 79.14 |
| Temperature Scaling | 89.24 | 61.67 | 76.62 | 78.52 |
| **Combined Method** | **89.68** | **62.86** | **77.35** | **80.00** |

It can be observed from Table 4 that the combined method yields a 1.6% performance improvement on CIFAR-10, a 4.9% performance improvement on CIFAR-100 and a 15.0% performance improvement on TinyImagenet compared to the baseline SimCLR. The improvement surpasses that achieved by using only margin tuning or temperature scaling. The combined method on the Mixed CIFAR-10 datasets also achieves performance improvements consistently as shown in Section A.5. The complete algorithm is presented in Algorithm 1.

**Alternative contrastive learning paradigm.** We delve deeper into the scalability of our methods across various self-supervised learning paradigms. Results in Table 5 demonstrate consistent performance enhancements comparable to those achieved by SimCLR on the MoCo on CIFAR-10.

**Complex classification scenarios.** We explore our method by targeting difficult samples under the long-tail classification scenario, where difficult samples are even more difficult to learn according to the imbalanced distributions. The findings in Table 6 illustrate that our approach outperforms the baseline SimCLR in scenarios involving distributional imbalance, indicating the adaptivity of our approach to complex classification scenarios.

Table 5: The results of incorporating the Combined method with different architectures on CIFAR-10.

| Method | MoCo | SimCLR |
|---|---|---|
| Baseline | 85.84 | 88.26 |
| Combined Method | **86.82** | **89.68** |

Table 6: Classification accuracy by using Combined method on TinyImagenet-LT. We also use SimCLR as the baseline method.

| Method | TinyImagenet-LT |
|---|---|
| Baseline | 43.34 |
| Combined Method | **47.62** |

**Further discussions.** We also provide a sensitivity analysis of parameters in Section A.4 and conduct a detailed analysis of results in Table 5 and Table 6 in Section A.5. Furthermore, discussions about which features are advantageous for selecting difficult examples are also presented in Section A.5. In Section A.5, we have also included the experimental results on ImageNet-1K, the trending of the derived bounds with Mixed CIFAR-10 dataset and the significance analysis of $\gamma$ and $\beta$.

## 6 CONCLUSION

In this paper, we construct a theoretical framework to specifically analyze the impact of difficult examples on contrastive learning. We prove that difficult examples hurt the performance of contrastive learning from the perspective of linear probing error bounds. We further demonstrate how techniques such as margin tuning, temperature scaling, and the removal of these examples from the dataset can improve performance from the perspective of enhancing the generalization bounds. The experimental results demonstrate the reliability of our theoretical analysis.

## ACKNOWLEDGEMENT

Yisen Wang was supported by National Natural Science Foundation of China (92370129, 62376010), Beijing Major Science and Technology Project under Contract no. Z251100008425006, Beijing Nova Program (20230484344, 20240484642), and State Key Laboratory of General Artificial Intelligence.

## ETHICS STATEMENT

Our research strictly adheres to ethical standards in the field of machine learning and artificial intelligence. This work makes use of publicly available datasets and models. No private or sensitive data are involved, and no harmful content is included. Therefore, we believe this paper does not raise any ethical concerns.

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

## A    Technical Appendices and Supplementary Material

### A.1    Related Works

**Self-supervised contrastive learning.** Self-supervised contrastive learning (Chen et al., 2020b;c; He et al., 2020; Wang et al., 2021a) aims to learn an encoder that maps augmentations (e.g. flips, random crops, etc.) of the same input to proximate features, while ensuring that augmentations of distinct inputs yield divergent features. The encoder, once pre-trained, is later fined-tuned on a specific downstream dataset. The effectiveness of contrastive learning methods are typically evaluated through the performances of the downstream tasks such as linear classification. Depending on the reliance of negative samples, contrastive learning methods can be broadly categorized into two kinds. The first kind (Chen et al., 2020b;c; He et al., 2020; Wang et al., 2024) learns the encoder by aligning an anchor point with its augmented versions (positive samples) while at the same time explicitly pushing away the others (negative samples). On the other hand, the second kind does not depend on negative samples (Zhuo et al., 2023). They often necessitate additional components like projectors (Grill et al., 2020; Ouyang et al., 2025), stop-gradient techniques (Chen & He, 2021), or high-dimensional embeddings (Zbontar et al., 2021). Nevertheless, the first kind of methods continue to be the mainstream in self-supervised contrastive learning and have been expanded into numerous other domains (Aberdam et al., 2021; Khaertdinov et al., 2021; Lee et al., 2022). The analysis and discussions of this paper focus mainly on the first kind of contrastive learning methods that relies on both positive and negative samples.

**Contrastive Learning Theory.** The early studies of theoretical aspects of contrastive learning manage to link contrastive learning to the supervised downstream classification. Arora et al. (2019) proves that representations learned by contrastive learning algorithms can achieve small errors in the downstream linear classification task. Ash et al. (2022); Bao et al. (2022); Nozawa & Sato (2021) incorporate the effect of negative samples and further extend surrogate bounds. Later on, HaoChen et al. (2021) focuses on the unsupervised nature of contrastive learning by modeling the feature similarities between augmented samples and provides generalization guarantee for linear evaluation through borrowing mathematical tools from spectral clustering. The idea of modeling similarities is later extended to analyzing contrastive learning for unsupervised domain adaption (Shen et al., 2022) and weakly supervised learning (Cui et al., 2023). In a similar vein, Wang et al. (2021b); Zhang et al. (2025) put forward the idea of *augmentation overlap* to explain the alignment of positive samples. Taking a step further, Cui et al. (2025) proposes an augmentation-aware error bound which enables the explanation of specific types of data augmentation. Besides, contrastive learning is also interpreted through various other theoretical frameworks in unsupervised learning, such as nonlinear independent component analysis (Zimmermann et al., 2021), neighborhood component analysis (Ko et al., 2022), stochastic neighbor embedding (Hu et al., 2023), geometric analysis of embedding spaces (Huang et al., 2023), and message passing techniques (Wang et al., 2023). In this paper, our basic assumptions are based on HaoChen et al. (2021) and focus on modeling the similarities between difficult example pairs.

**Difference between difficult examples and hard negative samples.** Difficult examples and hard negative samples both significantly affect the performance of self-supervised learning. However, while difficult examples are associated with the classification boundary, hard negative samples (Kalantidis et al., 2020; Robinson et al., 2020) are defined in relation to the anchor point. Previous research on hard negative sampling typically modifies contrastive learning models to emphasize these challenging samples so as to achieve better performance. In contrast, our findings indicate that unmodified contrastive learning models experience performance degradation due to the existence of difficult samples. Aside from ad hoc modifications, a straightforward removal of these difficult samples can also boost performance. As a systematic explanation of this finding is lacking, we establish a unified theoretical framework that addresses this challenge.

### A.2    Loss Functions of Sample Removal, Margin Tuning, and Temperature Scaling

Based on the sample selection matrix $P$ defined in equation 12, we adapt the InfoNCE loss into versions of sample removal, margin tuning, and temperature scaling, respectively.

**Sample Removal.** We define the removal loss as follows:

$$\ell_{\mathrm{R}}(i,j) := -\log \frac{\exp\big((s_{i,j}(1-p_{i,j}))/\tau\big)}{\sum_{k=1}^{2N} \mathbf{1}_{[k\neq i]} \exp\big((s_{i,k}(1-p_{i,k}))/\tau\big)}, \tag{13}$$

where $s_{i,j}$ denotes the similarity between augmented instances $x_i$ and $x_j$. If $p_{i,j}=0$, the sample pair $x_i$ and $x_j$ does not include difficult samples, so $(s_{i,j}(1-p_{i,j}))/\tau = s_{i,j}/\tau$, retaining the original form of the InfoNCE loss. If $p_{i,j}=1$, the sample pair $x_i$ and $x_j$ are difficult pairs, so $(s_{i,j}(1-p_{i,j}))/\tau = 0$, effectively removing them.

**Margin Tuning.** We start with the basic form of the widely used InfoNCE loss and define the margin tuning loss for each positive pair. Specifically, within each minibatch of size $N$, we generate $2N$ samples through data augmentation. Given the margin tuning factor $\sigma > 0$, for an anchor sample $x_i$ and its corresponding positive sample $x_j$, we define the margin tuning loss as follows:

$$\ell_{\mathrm{M}}(i,j) := -\log \frac{\exp\big((s_{i,j}+p_{i,j}\sigma)/\tau\big)}{\sum_{k=1}^{2N} \mathbf{1}_{[k\neq i]} \exp\big((s_{i,k}+p_{i,k}\sigma)/\tau\big)}, \tag{14}$$

where $s_{i,j}$ denotes the similarity between augmented instances $x_i$ and $x_j$, and $\tau > 0$ denotes the temperature parameter. After the above operation, we assign the same margin value to all selected difficult sample pairs, achieving the goal of margin tuning for specific sample pairs.

**Temperature Scaling.** To apply temperature scaling consistent with our theoretical analysis, we start with the basic form of the InfoNCE loss and define the temperature scaling loss for each positive pair. Specifically, within each minibatch, given the temperature scaling factor $\rho$, for an anchor sample $x_i$ and its corresponding positive sample $x_j$, we define the temperature scaling loss as follows:

$$\ell_{\mathrm{T}}(i,j) := -\log \frac{\exp\big(\frac{s_{i,j}}{[p_{i,j}\rho+(1-p_{i,j})]\tau}\big)}{\sum_{k=1}^{2N} \mathbf{1}_{[k\neq i]} \exp\big(\frac{s_{i,k}}{[p_{i,k}\rho+(1-p_{i,k})]\tau}\big)}, \tag{15}$$

where $s_{i,j}$ denotes the similarity between augmented instances $x_i$ and $x_j$.

**Combined Method.** The combined loss function as

$$\ell(i,j) := -\log \frac{\exp\big(\frac{s_{i,j}+p_{i,j}\sigma}{[p_{i,j}\rho+(1-p_{i,j})]\tau}\big)}{\sum_{k=1}^{2N} \mathbf{1}_{[k\neq i]} \exp\big(\frac{s_{i,k}+p_{i,k}\sigma}{[p_{i,k}\rho+(1-p_{i,k})]\tau}\big)}, \tag{16}$$

where $s_{i,j}$ denotes the similarity between augmented instances $x_i$ and $x_j$. The whole training procedure of the combined method is shown in Algorithm 1.

## A.3 TRAINING DETAILS

We run all experiments on an NVIDIA GeForce RTX 3090 24G GPU and we run experiments with ResNet-18 on the CIFAR-10, CIFAR-100 and STL-10 dataset and ResNet-50 on the TinyImagenet dataset. We only deal with the difficult examples during training time.

For CIFAR-10 we set batch size as 512, learning rate as 0.25 and base temperature as 0.5. We choose 0.15 as the $posHigh$ and 0.22 as the $posLow$. We set $\sigma$ as 0.03 and $\rho$ as 0.6 for CIFAR-10. For both our method and SimCLR, we evaluate the models using linear probing, when evaluating we set batch size as 512 and learning rate as 1. This experimental setup is also applicable to the Mixed CIFAR-10 dataset.

For CIFAR-100 we set batch size as 512, learning rate as 0.5 and base temperature as 0.1. We choose 0.013 as the $posHigh$ and 0.5 as the $posLow$. We set $\sigma$ as 0.1 and $\rho$ as 0.7 for CIFAR-100. For both our method and SimCLR, we evaluate the models using linear probing, when evaluating we set batch size as 512 and learning rate as 0.1.

For STL-10 we set batch size as 256, learning rate as 0.5 and base temperature as 0.1. We choose 0.15 as the $posHigh$ and 0.22 as the $posLow$. We set $\sigma$ as 0.1 and $\rho$ as 0.7 for STL-10. For both our method and SimCLR, we evaluate the models using linear probing, when evaluating we set batch size as 256 and learning rate as 0.1.

---

**Algorithm 1** Training procedure of Combined method

---

**Input:** batch size $N$, base temperature $\tau$, $posHigh$ and $posLow$ for determining the size of the interval, margin tuning factor $\sigma$, temperature scaling factor $\rho$, encoder $f(\cdot)$, projector $g(\cdot)$ and data augmentation $T$.

**Output:** encoder network $f(\cdot)$, and throw away $g(\cdot)$.

1: **for** sampled minibatch $\{\bar{x}_k\}_{k=1}^N$ **do**
2:     **for** all $k \in \{1,...,N\}$ **do**
3:         Draw two augmentation functions $t, t' \sim T$;
4:         $x_{2k-1} = t(\bar{x}_k)$ and $x_{2k} = t'(\bar{x}_k)$;
5:         $h_{2k-1} = f(x_{2k-1})$ and $h_{2k} = f(x_{2k})$;
6:         $z_{2k-1} = g(h_{2k-1})$ and $z_{2k} = g(h_{2k})$.
7:     **end for**
8:     **for** all $k \in \{1,...,2N\}$ **do**
9:         Calculate $P_i = (p_{i,j})_{j=1}^{2N}$ by using $h_j, j \in \{1,...,2N\}$ according to Eq. 12;
10:     **end for**
11:     The matrix $\boldsymbol{P}$ is obtained by splicing $P_i, i \in \{1,...,2N\}$ by rows.
12:     **for** all $i \in \{1,...,2N\}$ and all $j \in \{1,...,2N\}$ **do**
13:         $s_{i,j} = \boldsymbol{z}_i^\top \boldsymbol{z}_j / (\|\boldsymbol{z}_i\| \|\boldsymbol{z}_j\|)$.
14:     **end for**
15:     Calculate $\ell(i, j)$ according to Eq. 16;
16:     Calculate $\mathcal{L} = \frac{1}{2N} \sum_{k=1}^N [\ell(2k - 1, 2k) + \ell(2k, 2k - 1)]$; Update networks $f$ and $g$ to minimize $\mathcal{L}$.
17: **end for**

---

For TinyImagenet we set batch size as 512, learning rate as 0.5 and base temperature as 0.5. We choose 0.013 as the $posHigh$ and 0.5 as the $posLow$. We set $\sigma$ as 0.1 and $\rho$ as 0.7 for TinyImagenet. For both our method and SimCLR, we evaluate the models using linear probing, when evaluating we set batch size as 512 and learning rate as 0.1.

For the experimental results presented in Figure 1, we selected 20% SAS coreset for CIFAR-10, 95% SAS coreset for CIFAR-100, 80% SAS coreset for STL-10, and 60% SAS coreset for TinyImagenet, following the filtering method mentioned in (Joshi & Mirzasoleiman, 2023).

## A.4 PARAMETER SENSITIVITY ANALYSIS

**Evaluating different $\sigma$ used in margin tuning part**. The intention of $\sigma$ is to add margins to the similarity terms between difficult example pairs. We show the performance with different $\sigma$ in Figure 5(a), and the results show that when $\sigma = 0.1$ the proposal achieves the best performance on CIFAR-100, and the performance does not degrade significantly with $\sigma$ changes. This demonstrates that our proposal is quite robust with the selection of $\sigma$.

**Evaluating different $\rho$ used in temperature scaling part**. $\rho$ is used for scaling downwards the temperatures on the difficult example pairs so that we can eliminate the negative effects of difficult examples. We show the performance with different $\rho$ in Figure 5(b), and the results show that when $\rho = 0.7$ the proposal achieves the best performance on CIFAR-100, and the performance does not degrade significantly with $\rho$ changes. We figure out that different values of $\rho$ can all result in performance improvements.

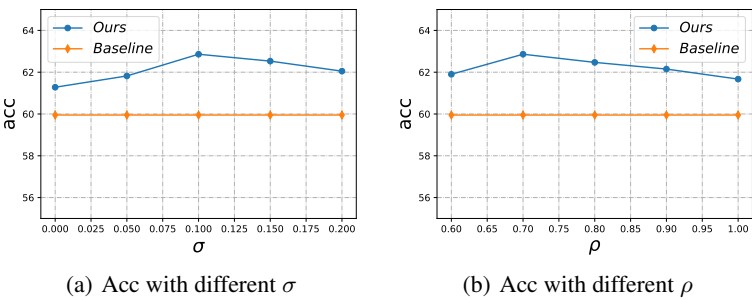

           (a) Acc with different $\sigma$            (b) Acc with different $\rho$

Figure 5: (a) Parameter analysis of margin tuning factor $\sigma$,(b) temperature scaling factor $\rho$, all of the above results are implemented on CIFAR-100.

A.5   FURTHER DISCUSSION

**Which feature is better for difficult examples selection?**   In SimCLR, the authors found that the proposal of projector $g(\cdot)$ allows the model to learn the auxiliary task better thus having better downstream generalization. However, as mentioned in (Cosentino et al., 2022) they suggest the problem of representation dimensional collapse after using projector, therefore, we here explore whether it is better to use features before projector $f(x)$ for difficult examples selection or $g(f(x))$ after projector.

Table 7: Classification accuracy by using Combined method on CIFAR-10 and CIFAR-100. Features before projector means that we use $f(x)$ for difficult examples selection and features after projector means that we use $g(f(x))$ for difficult examples selection.

| Features | Baseline | After projector | Before projector |
|---|---|---|---|
| CIFAR-10 | 88.26 | 87.86 | **89.68** |
| CIFAR-100 | 59.95 | 60.63 | **62.86** |

As shown in Table 7, We find that when using $f(x)$ rather than $g(f(x))$ for difficult examples selection we can gain a 2.1% performance improvement on CIFAR-10 and a 3.7% performance improvement on CIFAR-100. These results suggest that utilizing features before projector is more beneficial for difficult examples selection.

**The combined method is also effective for the Mixed CIFAR-10 datasets.**   As we discussed earlier, the Mixed CIFAR-10 datasets contain a large number of mixed difficult samples, making the learning difficulty of this dataset significantly greater than that of the original dataset. Based on this fact, this section explores whether our proposed method can achieve performance improvements on the Mixed CIFAR-10 datasets that are consistent with those on CIFAR-100, Tiny ImageNet, and other datasets. We use the 10%- and 20%-Mixed CIFAR-10 datasets as our baselines, while the 0%- Mixed CIFAR-10 datasets serve as our standard CIFAR-10 baseline.

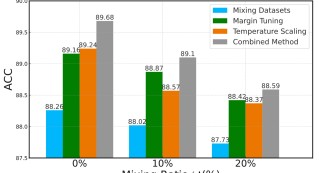

Figure 6: Detailed experimental results on the Mixed CI-FAR datasets.

The experimental results are shown in Figure 6. We found that using either margin tuning or temperature scaling alone can improve performance over the original baseline, while the combined method yields better results than using either approach individually. This finding is consistent with the experimental results on other datasets and further validates the effectiveness of our method.

**The proposal is effective for real-world datasets.**   We evaluated our method on the Imagenet-1k dataset, which contains 1,000 categories and 1,281,167 training samples. We used ResNet18 as our backbone, set the batch size to 1024, and resized each image to 96x96. We set the learning rate to 0.1 and the base temperature to 0.07. We chose 0.01 as the posHigh and 0.5 as the posLow. We set $\sigma$ to 0.1 and $\rho$ to 0.7. We also evaluated the models using linear probing. When evaluating, we set the batch size to 1024 and the learning rate to 1. The specific results are shown in Table 8.

Table 8: Classification accuracy on Imagenet-1k.

| Methods | Baseline | Removing | Temperature Scaling | Margin Tuning | Combined |
|---|---|---|---|---|---|
| Accuracy | 37.62 | 37.79 | 38.48 | 38.59 | 38.98 |

From the results on the real-world dataset, Imagenet-1k, which contains more categories, We can see that even after running for only 400 epochs, our method achieves a performance improvement trend consistent with the results mentioned in the paper, compared to the baseline method. These results strengthen the findings and demonstrate broader applicability of this paper.

**Focusing on difficult examples and removing them are both effective methods.**   We use temperature scaling as an example to illustrate how we should handle difficult examples. We note that placing greater emphasis on difficult examples (by selecting a smaller temperature) and discarding this sample (which is effectively equivalent to setting the temperature to infinity (we use a large value of 1,000,000,000 to approximate infinity here)) may seem contradictory. However, as shown

in Table 9, both approaches are indeed valid. This means that effectively handling difficult samples is possible under sufficiently good conditions, while in the absence of such mechanisms, simply discarding them can also be effective.

Table 9: Classification accuracy with various temperature scaling factors on CIFAR-100 datasets. Setting the Temperature Scaling Factor to 0.7 represents using our proposed theoretical framework to specifically address difficult samples, while setting the Temperature Scaling Factor to 1e9 means discarding these difficult samples. Results are averaged over three runs.

| Temperature Scaling Factor | 0.7 | 1 | 10 | 100 | 1000 | 1e9 |
|---|---|---|---|---|---|---|
| Accuracy | 61.67 | 59.95 | 59.63 | 59.82 | 60.05 | 60.31 |

**The scalability of our proposal under other contrastive learning paradigms.** As mentioned in (Johnson et al., 2022), InfoNCE and Spectral contrastive loss share the same population minimum with variant kernel derivations. By using similar techniques of positive-pair kernel, our conclusions can also be further generalized to other self-supervised learning frameworks. To demonstrate the scalability of the combined method, we supplement the comparative experiments based on the MoCo (Chen et al., 2020c) algorithm. The experimental results demonstrate consistent improvements of our method over both MoCo and SimCLR and show the scalability of our proposal under different contrastive learning paradigms.

**Connection between difficult examples and long-tailed distribution.** Under the definition that difficult examples contribute least to contrastive learning and that are consequently difficult to distinguish by contrastive learning models, we can easily draw the following conclusion: difficult examples can lead to unclear classification boundaries for the classes they belong to.

Due to the significant difference in the number of samples in the head and tail classes, the boundary of tail classes is difficult to be accurately estimated due to the tail classes are prone to collapse when the data is distributed with long-tailed distribution, as mentioned in (Samuel & Chechik, 2021). In other words, tail classes can lead to unclear classification boundaries for the classes they belong to as mentioned in (Fang et al., 2021).

So in this view, tail classes samples can also be seen as difficult samples. To better illustrate this point, we will further validate the connection between them through the following experiments. We validate our proposed Combined method on the classical long-tailed distribution dataset tiny-Imagenet-LT to explore whether our proposed algorithm can achieve a performance improvement over the comparison method SimCLR when distributional imbalance as another form of difficult samples also exists. The results in Table 6 show that we can achieve better performance when distributional imbalance also exists.

**Analysis of the trending of the derived bounds.** We analyze the trending of the derived bounds on the Mixed CIFAR-10 dataset. Specifically, we vary the mixing ratios from 0% to 30%, where 0% represents the standard CIFAR-10 without mixing. The experimental parameter settings can be referenced to A.3. For each class of samples, we sort them based on the difference between the maximum and second-largest values after applying softmax to the outputs, and select the 8% (the ratio is consistent with what is reported in the paper) smallest differences as the difficult examples, as described in the paper. For the calculation of $\alpha$, we take the mean of the similarity between all samples of the same class. For the calculation of $\beta$, we take the mean of the similarity for the sample pairs from different classes that do not contain the difficult examples. For the calculation of $\gamma$, we take the mean of the similarity for the sample pairs from different classes that contain the difficult examples.

In Table 10, we show that as the mixing ratio increases, the linear probing accuracy drops, and the $(K+1)$-th eigenvalue increases. Note that the classification error (left hand side of Eq.4) is 1-acc, and the error bound (right hand side of Eq.4) increases with the eigenvalue increasing. This result indicates that as the difficult examples increases, the classification error and the bound share the same variation trend, thus validating theorem 3.4 that larger $\gamma - \beta$ results in worse error bound.

**Significance analysis of $\gamma$ and $\beta$.** To verify the significance of $\gamma$ and $\beta$., we tested $\gamma$ and $\beta$, as well as $\gamma - \beta$, on more real datasets. From the first three rows of Table 11, we found that on the CIFAR-100 dataset (which has 10 times more classes than CIFAR-10), the difference between $\gamma$ and $\beta$ remained consistent with that on the CIFAR-10 dataset. On the ImageNet-1k dataset (which has 100

Table 10: The trends of $\alpha$, $\beta$, $\gamma$, and other metrics as the Mixing Ratio changes.

| Mixing Ratio | 0% | 10% | 20% | 30% |
|---|---|---|---|---|
| acc (%) | 88.3 | 88.0 | 87.7 | 86.2 |
| $\alpha$ | 47.2 | 44.0 | 41.2 | 38.7 |
| $\beta$ | 19.1 | 19.5 | 20.1 | 20.8 |
| $\gamma$ | 20.9 | 22.1 | 23.1 | 24.1 |
| $\gamma - \beta$ | 1.80 | 2.60 | 3.00 | 3.30 |
| Eigenvalue ($\times 10^{-5}$) | 2.93 | 3.36 | 3.58 | 3.72 |

times more classes than CIFAR-10,for specific experimental details and results on ImageNet-1k), the difference between $\gamma$ and $\beta$ was even larger than on CIFAR-10. As a possible intuitive explanation, we conjecture that the higher $\gamma - \beta$ might results from the higher complexity of imagenet images, e.g. different-class images with similar backgrounds can share higher similarity (higher $\gamma$), whereas CIFAR images have relatively simple and consistent backgrounds. These results demonstrate that even on real-world datasets, the difference between $\gamma$ and $\beta$ is significant.

Table 11: Comparison of $\beta$, $\gamma$, $\gamma - \beta$ , t-statistic and P value across different datasets.

| Datasets | CIFAR-10 | CIFAR-100 | Imagenet-1k |
|---|---|---|---|
| $\beta$ | 19.1 | 35.6 | 39.8 |
| $\gamma$ | 20.9 | 37.4 | 42.9 |
| $\gamma - \beta$ | 1.8 | 1.8 | 3.1 |
| $t$-statistic | -502.63 | -539.36 | -3844.21 |
| $P$ value | 0.0 | 0.0 | 0.0 |

To better illustrate the significant difference between $\gamma$ and $\beta$, we conducted an independent samples t-test to support our conclusion. Specifically, we first collected all the $\beta$ and $\gamma$ values, and due to the large sample size, we chose to use Welch's t-test, which does not assume equal variances between the two groups and is suitable for cases where the variances may differ. In the experiment, we focus on two key statistics:

t-statistic: This measures the difference between the means of the two groups relative to the variance within the samples. The t-statistic is a standardized measure used to determine whether the mean difference between the two groups is significant or could be attributed to random fluctuations. The larger the t-statistic, the more significant the difference between the two groups.

P value: The p-value indicates the probability of observing the current difference or more extreme results under the assumption that the null hypothesis (i.e., no significant difference between the two groups) is true. If the p-value is less than 0.05, it suggests that the observed difference is highly unlikely under the null hypothesis, and we can reject the null hypothesis, concluding that there is a significant difference between the two groups.

As shown in the last two rows of Table 11, on all datasets (CIFAR-10, CIFAR-100, Imagenet-1k), the absolute value of the T-statistic is very large, and the P-value is close to zero. This indicates that the mean difference between $\gamma$ and $\beta$ is highly statistically significant.

## B    PROOFS

Recall that in Section 3.2, we introduce the adjacency matrix of the similarity graph based on a 4-sample subset. Here we give the formal definition of the adjacency matrix of a generalized similarity graph containing $|\mathcal{X}| = n(r + 1)$ samples, with $n$ denoting the number of augmented samples per class, and $r + 1$ denoting the number of classes.

Denote $\mathbb{D} = \{x_1, \ldots, x_{n(r+1)}\}$ as the dataset, where $x_{n(i-1)+1}, \ldots, x_{ni}$ belong to Class $i$ for $i \in \{1, \ldots, r+1\}$. Denote $n_d$ as the number of difficult examples per class and $\mathbb{D}_d$ as the set of difficult examples. Naturally, we denote $n_e := n - n_d$ as the number of easy-to-learn examples per class. Without loss of generality, we assume that the last $n_d$ examples in each class are difficult examples. Moreover, as we define difficult examples in a pairwise manner, without loss of generality, we

assume that $x_{n(i-1)+l}$ and $x_{n(j-1)+l}$ are a pair of difficult examples for $i \neq j \in \{1, \ldots, r+1\}$ and $l \in \{n_e + 1, \ldots, n\}$. Let $\beta < \gamma < \alpha < 1$. Then we define the elements of the adjacency matrix $\boldsymbol{A} = (w_{x,x'})_{x,x' \in \mathcal{X}}$ as $w_{x,x'} := 1$ for $x = x'$; $w_{x,x'} := \alpha$ for $x \neq x'$, $y(x) = y(x')$; $w_{x,x'} := \gamma$ for $x, x' \in \mathbb{D}_d$, $y(x) \neq y(x')$, and $w_{x,x'} := \beta$ otherwise.

Specifically, we have the adjacency matrix of a similarity graph without difficult examples as

$$
\boldsymbol{A}_{\text{w.o.}} = \begin{bmatrix} \boldsymbol{A}_{\text{same-class}} & \boldsymbol{A}_{\text{different-class}} & \cdots & \boldsymbol{A}_{\text{different-class}} \\ \boldsymbol{A}_{\text{different-class}} & \boldsymbol{A}_{\text{same-class}} & \cdots & \boldsymbol{A}_{\text{different-class}} \\ \vdots & \vdots & & \vdots \\ \boldsymbol{A}_{\text{different-class}} & \boldsymbol{A}_{\text{different-class}} & \cdots & \boldsymbol{A}_{\text{same-class}} \end{bmatrix}_{(r+1) \times (r+1)} \tag{17}
$$

and the adjacency matrix of a similarity graph with difficult examples as

$$
\boldsymbol{A}_{\text{w.d.}} = \begin{bmatrix} \boldsymbol{A}_{\text{same-class}} & \boldsymbol{A}'_{\text{different-class}} & \cdots & \boldsymbol{A}'_{\text{different-class}} \\ \boldsymbol{A}'_{\text{different-class}} & \boldsymbol{A}_{\text{same-class}} & \cdots & \boldsymbol{A}'_{\text{different-class}} \\ \vdots & \vdots & & \vdots \\ \boldsymbol{A}'_{\text{different-class}} & \boldsymbol{A}'_{\text{different-class}} & \cdots & \boldsymbol{A}_{\text{same-class}} \end{bmatrix}_{(r+1) \times (r+1)} \tag{18}
$$

where

$$
\boldsymbol{A}_{\text{same-class}} = \begin{bmatrix} 1 & \alpha & \cdots & \alpha \\ \alpha & 1 & \cdots & \alpha \\ \cdots & & & \\ \alpha & \alpha & \cdots & 1 \end{bmatrix}_{n \times n}, \tag{19}
$$

$$
\boldsymbol{A}_{\text{different-class}} = \begin{bmatrix} \beta & \cdots & \beta \\ \vdots & & \vdots \\ \beta & \cdots & \beta \end{bmatrix}_{n \times n}, \tag{20}
$$

and

$$
\boldsymbol{A}'_{\text{different-class}} = \begin{bmatrix} \beta & \cdots & \beta & \beta & \cdots & \beta \\ \vdots & & \vdots & \vdots & & \vdots \\ \beta & \cdots & \beta & \beta & \cdots & \beta \\ \beta & \cdots & \beta & \gamma & \cdots & \beta \\ \vdots & & \vdots & \vdots & \ddots & \vdots \\ \beta & \cdots & \beta & \beta & \cdots & \gamma \end{bmatrix}_{(n_e + n_d) \times (n_e + n_d)}. \tag{21}
$$

### B.1 PROOFS RELATED TO SECTION 3.3

*Proof of Theorem 3.3.* For a dataset without difficult examples, the similarity between a sample and itself is 1, the similarity between same-class samples is $\alpha$, and the similarity between different-class samples is $\beta$. Then the adjacent matrix of the similarity graph can be decomposed into the sum of several matrix Kronecker products:

$$
\boldsymbol{A} = (1 - \alpha)\boldsymbol{I}_{r+1} \otimes \boldsymbol{I}_n + (\alpha - \beta)\boldsymbol{I}_{r+1} \otimes (\boldsymbol{1}_n \cdot \boldsymbol{1}_n^\top) + \beta(\boldsymbol{1}_{r+1} \cdot \boldsymbol{1}_{r+1}^\top) \otimes (\boldsymbol{1}_n \cdot \boldsymbol{1}_n^\top), \tag{22}
$$

where $\boldsymbol{I}_{r+1}$ and $\boldsymbol{I}_n$ denote the $(r+1) \times (r+1)$ and $n \times n$ identity matrices respectively, and $\boldsymbol{1}_{r+1} := (1, \ldots, 1)^\top \in \mathbb{R}^{r+1}$ and $\boldsymbol{1}_n := (1, \ldots, 1)^\top \in \mathbb{R}^n$ denote the all-one vectors.

First, we calculate the eigenvalues and eigenvectors of $\boldsymbol{A}$. Note that $\boldsymbol{I}_{r+1}$ and $\boldsymbol{I}_n$ have eigenvalues 1 with arbitrary eigenvectors, $\boldsymbol{1}_n \cdot \boldsymbol{1}_n^\top$ has eigenvalue $n$ with eigenvector $\bar{\boldsymbol{1}}_n := \frac{1}{\sqrt{n}}\boldsymbol{1}_n$ and eigenvalues 0 with eigenvectors $\{\mu : \mu^\top \boldsymbol{1}_n = 0\}$, and $\boldsymbol{1}_{r+1} \cdot \boldsymbol{1}_{r+1}^\top$ has eigenvalue $r+1$ with eigenvector $\bar{\boldsymbol{1}}_{r+1} := \frac{1}{\sqrt{r+1}}\boldsymbol{1}_{r+1}$ and eigenvalues 0 with eigenvectors $\{\nu : \nu^\top \boldsymbol{1}_{r+1} = 0\}$. Therefore, $\boldsymbol{A}$ has the following sets of eigenvalues and eigenvectors:

$$
\lambda_1 = (1 - \alpha) + n(\alpha - \beta) + n(r+1)\beta, \qquad \text{with eigenvector } \bar{\boldsymbol{1}}_{r+1} \otimes \bar{\boldsymbol{1}}_n;
$$

$$\lambda_2 = \ldots = \lambda_{r+1} = (1-\alpha) + n(\alpha - \beta), \qquad \text{with eigenvectors } \nu \otimes \bar{\mathbf{1}}_n;$$
$$\lambda_{r+2} = \ldots = \lambda_{n+r} = 1 - \alpha, \qquad \text{with eigenvectors } \bar{\mathbf{1}}_{r+1} \otimes u;$$
$$\lambda_{n+r+1} = \ldots = \lambda_{n(r+1)} = 1 - \alpha, \qquad \text{with eigenvectors } u \otimes v.$$

Next, we calculate the eigenvalues of $\bar{\boldsymbol{A}} := \boldsymbol{D}^{-1/2}\boldsymbol{A}\boldsymbol{D}^{-1/2}$. By definition, we have $\boldsymbol{D} = \mathrm{diag}(w_1, \ldots, w_{n(r+1)}) = [(1-\alpha) + n\alpha + nr\beta]\boldsymbol{I}_{n(r+1)}$. Therefore, we have the eigenvalues of $\boldsymbol{A}$ as

$$\lambda_1 = 1,$$
$$\lambda_2 = \ldots = \lambda_{r+1} = \frac{(1-\alpha) + n(\alpha - \beta)}{(1-\alpha) + n\alpha + nr\beta},$$
$$\lambda_{r+2} = \ldots = \lambda_{n(r+1)} = \frac{1-\alpha}{(1-\alpha) + n\alpha + nr\beta}.$$

Then according to Theorem B.3 in HaoChen et al. (2021), when $k > r$, there holds

$$\mathcal{E}_{\text{w.o.}} \leq \frac{4\delta}{1 - \lambda_{k+1}} + 8\delta = \frac{4\delta}{1 - \frac{1-\alpha}{(1-\alpha)+n\alpha+nr\beta}} + 8\delta. \tag{23}$$

$\square$

*Proof of Theorem 3.4.* For a dataset with $n_d$ difficult examples per class, the similarity between a sample and itself is 1, the similarity between same-class samples is $\alpha$, the similarity between different-class easy-to-learn samples is $\beta$, and the similarity between different-class hard-to-learn samples is $\gamma$. Without loss of generality, we assume that $n$ is an integral multiple of $n_d$, i.e. there exist a $\kappa \in \mathbb{Z}^+$ such that $n = \kappa n_d$. Then the adjacent matrix of the similarity graph can be decomposed into the sum of several matrix Kronecker products:

$$\boldsymbol{A} = (1-\alpha)\boldsymbol{I}_{r+1} \otimes \boldsymbol{I}_n + (\alpha - \beta)\boldsymbol{I}_{r+1} \otimes (\mathbf{1}_n \cdot \mathbf{1}_n^\top) + \beta(\mathbf{1}_{r+1} \cdot \mathbf{1}_{r+1}^\top) \otimes (\mathbf{1}_n \cdot \mathbf{1}_n^\top)$$
$$+ (\gamma - \beta)(\mathbf{1}_{r+1} \cdot \mathbf{1}_{r+1}^\top) \otimes (\boldsymbol{e}_\kappa \cdot \boldsymbol{e}_\kappa^\top) \otimes \boldsymbol{I}_{n_d} - (\gamma - \beta)\boldsymbol{I}_{r+1} \otimes (\boldsymbol{e}_\kappa \cdot \boldsymbol{e}_\kappa^\top) \otimes \boldsymbol{I}_{n_d}, \tag{24}$$

where $\boldsymbol{I}_{r+1}$, $\boldsymbol{I}_n$, and $\boldsymbol{I}_{n_d}$ denote the $(r+1) \times (r+1)$, $n \times n$, and $n_d \times n_d$ identity matrices respectively, $\mathbf{1}_{r+1} := (1, \ldots, 1)^\top \in \mathbb{R}^{r+1}$ and $\mathbf{1}_n := (1, \ldots, 1)^\top \in \mathbb{R}^n$ denote the all-one vectors, and $\boldsymbol{e}_\kappa := (0, \ldots, 0, 1)^\top \in \mathbb{R}^\kappa$.

Similarly, we can decompose $\boldsymbol{D}$ into

$$\boldsymbol{D} = \boldsymbol{I}_{r+1} \otimes \Big[ [(1-\alpha) + n\alpha + nr\beta]\boldsymbol{I}_n + r(\gamma - \beta)(\boldsymbol{e}_\kappa \cdot \boldsymbol{e}_\kappa^\top) \otimes \boldsymbol{I}_{n_d} \Big], \tag{25}$$

and therefore we have

$$\boldsymbol{D}^{-1} = \boldsymbol{I}_{r+1} \otimes \Big[ \frac{1}{c_2}[\boldsymbol{I}_\kappa - (\boldsymbol{e}_\kappa \cdot \boldsymbol{e}_\kappa^\top)] + \frac{1}{c_1}(\boldsymbol{e}_\kappa \cdot \boldsymbol{e}_\kappa^\top) \Big] \otimes \boldsymbol{I}_{n_d}, \tag{26}$$

where we denote $c_1 := (1-\alpha) + n\alpha + nr\beta + nr(\gamma - \beta)$ and $c_2 := (1-\alpha) + n\alpha + nr\beta$.

Then we have the decomposition of the normalized similarity matrix as

$$\bar{\boldsymbol{A}} = \boldsymbol{D}^{-1/2}\boldsymbol{A}\boldsymbol{D}{-1/2}$$

$$= (1-\alpha)\boldsymbol{I}_{r+1} \otimes \Big[ \frac{1}{c_2}[\boldsymbol{I}_\kappa - (\boldsymbol{e}_\kappa \cdot \boldsymbol{e}_\kappa^\top)] + \frac{1}{c_1}(\boldsymbol{e}_\kappa \cdot \boldsymbol{e}_\kappa^\top) \Big] \otimes \boldsymbol{I}_{n_d}$$

$$+ (\gamma - \beta)(\mathbf{1}_{r+1} \cdot \mathbf{1}_{r+1}^\top) \otimes \frac{1}{c_1}(\boldsymbol{e}_\kappa \cdot \boldsymbol{e}_\kappa^\top) \otimes \boldsymbol{I}_{n_d}$$

$$- (\gamma - \beta)\boldsymbol{I}_{r+1} \otimes \frac{1}{c_1}(\boldsymbol{e}_\kappa \cdot \boldsymbol{e}_\kappa^\top) \otimes \boldsymbol{I}_{n_d}.$$

$$+ (\alpha - \beta)\boldsymbol{I}_{r+1} \otimes \Big[ [\frac{1}{\sqrt{c_2}}(\mathbf{1}_\kappa - \boldsymbol{e}_\kappa) + \frac{1}{\sqrt{c_1}}\boldsymbol{e}_\kappa] \cdot [\frac{1}{\sqrt{c_2}}(\mathbf{1}_\kappa - \boldsymbol{e}_\kappa) + \frac{1}{\sqrt{c_1}}\boldsymbol{e}_\kappa]^\top \Big] \otimes (\mathbf{1}_{n_d} \cdot \mathbf{1}_{n_d}^\top)$$

$$+ \beta(\mathbf{1}_{r+1} \cdot \mathbf{1}_{r+1}^\top) \otimes \Big[ [\frac{1}{\sqrt{c_2}}(\mathbf{1}_\kappa - \boldsymbol{e}_\kappa) + \frac{1}{\sqrt{c_1}}\boldsymbol{e}_\kappa] \cdot [\frac{1}{\sqrt{c_2}}(\mathbf{1}_\kappa - \boldsymbol{e}_\kappa) + \frac{1}{\sqrt{c_1}}\boldsymbol{e}_\kappa]^\top \Big] \otimes (\mathbf{1}_{n_d} \cdot \mathbf{1}_{n_d}^\top)$$

$$
= \frac{1}{c_2}(1-\alpha)\boldsymbol{I}_{r+1} \otimes \boldsymbol{I}_\kappa \otimes \boldsymbol{I}_{n_d}
$$
$$
+ \frac{1}{c_1}(\gamma-\beta)(\mathbf{1}_{r+1} \cdot \mathbf{1}_{r+1}^\top) \otimes (\boldsymbol{e}_\kappa \cdot \boldsymbol{e}_\kappa^\top) \otimes \boldsymbol{I}_{n_d}
$$
$$
- \Big[\frac{1}{c_1}(\gamma-\beta) + (\frac{1}{c_2}-\frac{1}{c_1})(1-\alpha)\Big]\boldsymbol{I}_{r+1} \otimes (\boldsymbol{e}_\kappa \cdot \boldsymbol{e}_\kappa^\top) \otimes \boldsymbol{I}_{n_d}.
$$
$$
+ (\alpha-\beta)\boldsymbol{I}_{r+1} \otimes \Big[[\frac{1}{\sqrt{c_2}}(\mathbf{1}_\kappa - \boldsymbol{e}_\kappa) + \frac{1}{\sqrt{c_1}}\boldsymbol{e}_\kappa] \cdot [\frac{1}{\sqrt{c_2}}(\mathbf{1}_\kappa - \boldsymbol{e}_\kappa) + \frac{1}{\sqrt{c_1}}\boldsymbol{e}_\kappa]^\top\Big] \otimes (\mathbf{1}_{n_d} \cdot \mathbf{1}_{n_d}^\top)
$$
$$
+ \beta(\mathbf{1}_{r+1} \cdot \mathbf{1}_{r+1}^\top) \otimes \Big[[\frac{1}{\sqrt{c_2}}(\mathbf{1}_\kappa - \boldsymbol{e}_\kappa) + \frac{1}{\sqrt{c_1}}\boldsymbol{e}_\kappa] \cdot [\frac{1}{\sqrt{c_2}}(\mathbf{1}_\kappa - \boldsymbol{e}_\kappa) + \frac{1}{\sqrt{c_1}}\boldsymbol{e}_\kappa]^\top\Big] \otimes (\mathbf{1}_{n_d} \cdot \mathbf{1}_{n_d}^\top).
\tag{27}
$$

Now we calculate the eigenvalues and eigenvectors of $\boldsymbol{A}$. For notational simplicity, we denote the first three terms of equation 27 as $\bar{\boldsymbol{A}}_1$ and the last two terms as $\bar{\boldsymbol{A}}_2$. Note that $\boldsymbol{I}_{r+1}, \boldsymbol{I}_\kappa$, and $\boldsymbol{I}_{n_d}$ have eigenvalues 1 with arbitrary eigenvectors, $\mathbf{1}_{r+1} \cdot \mathbf{1}_{r+1}^\top$ has eigenvalue $r+1$ with eigenvector $\bar{\mathbf{1}}_{r+1} := \frac{1}{\sqrt{r+1}}\mathbf{1}_{r+1}$ and eigenvalues 0 with eigenvectors $\{\nu : \nu^\top \mathbf{1}_{r+1} = 0\}$, and $\boldsymbol{e}_\kappa \cdot \boldsymbol{e}_\kappa^\top$ has eigenvalue 1 with eigenvector $\boldsymbol{e}_1 = (1,0,\ldots,0)^\top \in \mathbb{R}^\kappa$ and eigenvalues 0 with eigenvectors $\{\boldsymbol{e}_2,\ldots,\boldsymbol{e}_\kappa\}$. Let $\xi \in \mathbb{R}^{n_d}$ denote an arbitrary vector. Then $\bar{\boldsymbol{A}}_1$ has the following sets of eigenvalues and eigenvectors:

$$
\lambda_{1,1} = \ldots = \lambda_{1,n_d} = \frac{1}{c_2}(1-\alpha) + \frac{1}{c_1}(\gamma-\beta)(r+1) - \Big[\frac{1}{c_1}(\gamma-\beta) + (\frac{1}{c_2}-\frac{1}{c_1})(1-\alpha)\Big],
$$
$$
= \frac{1}{c_1}(1-\alpha) + \frac{r}{c_1}(\gamma-\beta), \text{ with eigenvectors } \bar{\mathbf{1}}_{r+1} \otimes \boldsymbol{e}_1 \otimes \xi;
$$
$$
\lambda_{1,n_d+1} = \ldots = \lambda_{1,n} = \frac{1}{c_2}(1-\alpha), \text{ with eigenvectors } \bar{\mathbf{1}}_{r+1} \otimes \boldsymbol{e}_i \otimes \xi, i = 2,\ldots,\kappa;
$$
$$
\lambda_{1,n+1} = \ldots = \lambda_{1,(r+1)n-rn_d} = \frac{1}{c_2}(1-\alpha), \text{ with eigenvectors } \nu \otimes \boldsymbol{e}_i \otimes \xi, i = 2,\ldots,\kappa;
$$
$$
\lambda_{1,(r+1)n-rn_d+1} = \ldots = \lambda_{1,(r+1)n} = \frac{1}{c_2}(1-\alpha) - \Big[\frac{1}{c_1}(\gamma-\beta) + (\frac{1}{c_2}-\frac{1}{c_1})(1-\alpha)\Big],
$$
$$
= \frac{1}{c_1}(1-\alpha) - \frac{1}{c_1}(\gamma-\beta), \text{ with eigenvectors } \nu \otimes \boldsymbol{e}_1 \otimes \xi.
$$

On the other hand, note that $\mathbf{1}_{n_d} \cdot \mathbf{1}_{n_d}^\top$ has eigenvalue $n_d$ with eigenvector $\bar{\mathbf{1}}_{n_d} := \frac{1}{\sqrt{n_d}}\mathbf{1}_{n_d}$ and eigenvalues 0 with eigenvectors $\{\eta : \eta^\top \mathbf{1}_{n_d} = 0\}$, and that by calculations, $[\frac{1}{\sqrt{c_2}}(\mathbf{1}_\kappa - \boldsymbol{e}_\kappa) + \frac{1}{\sqrt{c_1}}\boldsymbol{e}_\kappa] \cdot [\frac{1}{\sqrt{c_2}}(\mathbf{1}_\kappa - \boldsymbol{e}_\kappa) + \frac{1}{\sqrt{c_1}}\boldsymbol{e}_\kappa]^\top$ has eigenvalue $\frac{\kappa-1}{c_2} + \frac{1}{c_1}$ with eigenvector $\{\eta : \sum_{i=1}^{\kappa-1}\eta_i = 0, \eta_\kappa = (\kappa-1)\sqrt{c_1/c_2}\}$ and eigenvalues 0 with eigenvectors $\{\theta : \sum_{i=1}^{\kappa-1}\theta_i = 0, \eta_\kappa = 0\}$. Then $\bar{\boldsymbol{A}}_2$ has the following sets of eigenvalues and eigenvectors:

$$
\lambda_{2,1} = (\alpha-\beta)\Big[\frac{\kappa-1}{c_2} + \frac{1}{c_1}\Big]n_d + \beta(r+1)\Big[\frac{\kappa-1}{c_2} + \frac{1}{c_1}\Big]n_d,
$$
$$
= (\alpha+r\beta)\Big[\frac{\kappa-1}{c_2} + \frac{1}{c_1}\Big]n_d, \text{ with eigenvectors } \bar{\mathbf{1}}_{r+1} \otimes \eta \otimes \bar{\mathbf{1}}_{n_d};
$$
$$
\lambda_{2,2} = \ldots = \lambda_{2,r+1} = (\alpha-\beta)\Big[\frac{\kappa-1}{c_2} + \frac{1}{c_1}\Big]n_d, \text{ with eigenvectors } \nu \otimes \eta \otimes \bar{\mathbf{1}}_{n_d};
$$
$$
\lambda_{2,r+2} = \ldots = \lambda_{2,(r+1)n} = 0, \text{ with other combinations of eigenvectors.}
$$

By Equation 13 in Fulton (2000), for two real symmetric $n(r+1) \times n(r+1)$ matrices $\bar{\boldsymbol{A}}_1$ and $\bar{\boldsymbol{A}}_2$, we have the $k+1$-th largest eigenvalue of $\bar{\boldsymbol{A}} := \bar{\boldsymbol{A}}_1 + \bar{\boldsymbol{A}}_2$ satisfies

$$
\lambda_{k+1} \le \min_{i+j=k+2} \lambda_{1,i} + \lambda_{2,j}
$$

$$= \begin{cases} \dfrac{1}{c_1}(1-\alpha) + \dfrac{r}{c_1}(\gamma-\beta) + (\alpha-\beta)\Big[\dfrac{\kappa-1}{c_2} + \dfrac{1}{c_1}\Big]n_d, & \text{for } k < r+1, \\[2ex] \min\Big\{\dfrac{1}{c_1}(1-\alpha) + \dfrac{r}{c_1}(\gamma-\beta), \dfrac{1}{c_2}(1-\alpha) + (\alpha-\beta)\Big[\dfrac{\kappa-1}{c_2} + \dfrac{1}{c_1}\Big]n_d\Big\} \\[2ex] \quad = \dfrac{1}{c_1}(1-\alpha) + \dfrac{r}{c_1}(\gamma-\beta), & \text{for } r+1 \le k < n_d+r+1, \\[2ex] \quad = \dfrac{1}{c_2}(1-\alpha), & \text{for } n_d+r+1 \le k < (r+1)n - n_d r + r + 1. \end{cases}$$

Then according to Theorem B.3 in HaoChen et al. (2021), when $r+1 \le k < n_d+r+1$, there holds

$$\mathcal{E}_{\text{w.d.}} \le \frac{4\delta}{1-\lambda_{k+1}} + 8\delta = \frac{4\delta}{1 - \frac{1}{c_1}(1-\alpha) - \frac{r}{c_1}(\gamma-\beta)} + 8\delta = \frac{4\delta}{1 - \frac{(1-\alpha)+r(\gamma-\beta)}{(1-\alpha)+n\alpha+nr\beta+r(\gamma-\beta)}} + 8\delta. \tag{28}$$

$\square$

## B.2 Proofs Related to Section 4

*Proof of Corollary 4.1.* By removing the difficult examples, we have the adjacency matrix as

$$A = \begin{bmatrix} A_{\text{same}-\text{class}} & A_{\text{different}-\text{class}} & \cdots & A_{\text{different}-\text{class}} \\ A_{\text{different}-\text{class}} & A_{\text{same}-\text{class}} & \cdots & A_{\text{different}-\text{class}} \\ \vdots & \vdots & & \vdots \\ A_{\text{different}-\text{class}} & A_{\text{different}-\text{class}} & \cdots & A_{\text{same}-\text{class}} \end{bmatrix}_{(r+1)\times(r+1)}, \tag{29}$$

where

$$A_{\text{different}-\text{class}} = \begin{bmatrix} \beta & \cdots & \beta \\ \vdots & & \vdots \\ \beta & \cdots & \beta \end{bmatrix}_{n_e \times n_e}. \tag{30}$$

Then the matrix $A$ reduces to $A_{\text{w.o.}}$ and the error bound reduces to that in Theorem 3.3 with $n$ replaced with $n_e = n - n_d$. $\square$

The spectral contrastive loss with a margin $M = (m_{x,x'})$ is defined as

$$\mathcal{L}_M(\boldsymbol{x}; f) = -2\mathbb{E}_{x,x^+} f(x)^\top f(x^+) + \mathbb{E}_{x,x'}\Big[f(x)^\top f(x') + m_{x,x'}\Big]^2. \tag{31}$$

*Proof of Theorem 4.2.*

$$\begin{aligned} \mathcal{L}_M &= -2\mathbb{E}_{x,x^+} f(x)^\top f(x^+) + \mathbb{E}_{x,x'}\Big[f(x)^\top f(x') + m_{x,x'}\Big]^2 \\ &= -2\sum_{x,x^+} w_{x,x^+} f(x)^\top f(x^+) + \sum_{x,x'} w_x w_{x'}\Big[f(x)^\top f(x') + m_{x,x'}\Big]^2 \\ &= \sum_{x,x'} \Big\{ -2w_{x,x'} f(x)^\top f(x') + w_x w_{x'}\Big[f(x)^\top f(x')\Big]^2 + 2w_x w_{x'} m_{x,x'} f(x)^\top f(x') + w_x w_{x'} m_{x,x'}^2 \Big\} \\ &= \sum_{x,x'} \Big\{ w_x w_{x'}\Big[f(x)^\top f(x')\Big]^2 - 2[w_{x,x'} - w_x w_{x'} m_{x,x'}]f(x)^\top f(x') + w_x w_{x'} m_{x,x'}^2 \Big\} \\ &= \sum_{x,x'} \Big\{ \Big[[\sqrt{w_x} f(x)]^\top [\sqrt{w_{x'}} f(x')]\Big]^2 - 2\Big[\frac{w_{x,x'}}{\sqrt{w_x}\sqrt{w_{x'}}} - \sqrt{w_x}\sqrt{w_{x'}} m_{x,x'}\Big][\sqrt{w_x} f(x)]^\top [\sqrt{w_{x'}} f(x')] \\ &\qquad + \Big[\frac{w_{x,x'}}{\sqrt{w_x}\sqrt{w_{x'}}} - \sqrt{w_x}\sqrt{w_{x'}} m_{x,x'}\Big]^2 + 2w_{x,x'} m_{x,x'} - \frac{w_{x,x'}^2}{w_x w_{x'}} \Big\} \\ &= \sum_{x,x'} \Big[\frac{w_{x,x'}}{\sqrt{w_x}\sqrt{w_{x'}}} - \sqrt{w_x}\sqrt{w_{x'}} m_{x,x'} - [\sqrt{w_x} f(x)]^\top [\sqrt{w_{x'}} f(x')]\Big]^2 + \sum_{x,x'}\Big(2w_{x,x'} m_{x,x'} - \frac{w_{x,x'}^2}{w_x w_{x'}}\Big) \end{aligned}$$

$$:= \|(\bar{A} - \bar{M}) - FF^\top\|_F^2 + \sum_{x,x'} \left( 2w_{x,x'} m_{x,x'} - \frac{w_{x,x'}^2}{w_x w_{x'}} \right), \tag{32}$$

where we denote $\bar{A} := D^{-1/2} A D^{-1/2}$, $\bar{M} := D^{1/2} M D^{1/2}$, $A := (w_{x,x'})_{x,x' \in \{x_i\}_{i=1}^{n(r+1)}}$, $M := (m_{x,x'})_{x,x' \in \{x_i\}_{i=1}^{n(r+1)}}$, $D := \mathrm{diag}(w_1, \ldots, w_{n(r+1)})$, and $F = (\sqrt{w_x} f(x))_{x \in \{x_i\}_{i=1}^{n(r+1)}}$.

Note that given the adjacency matrix of the similarity graph $A$ and the margin matrix $M$, the second term in equation 32 is a constant. Therefore, minimizing the margin tuning loss $\mathcal{L}_{\mathrm{M}}$ over $f(x)$ is equivalent to minimizing the matrix factorization loss $\mathcal{L}_{\mathrm{mf}-\mathrm{M}} := \|(\bar{A} - \bar{M}) - FF^\top\|_F^2$ over $F$. $\square$

*Proof of Theorem 4.3.* Recall that when difficult examples exist, we assume that

$$w_{x,x'} := \begin{cases} 1 & \text{for } x = x', \\ \alpha & \text{for } x \neq x', y(x) = y(x'), \\ \gamma & \text{for } x, x' \in \mathbb{D}_d, y(x) \neq y(x'), \\ \beta & \text{otherwise.} \end{cases} \tag{33}$$

Then by definition we have

$$w_x = \sum_{x'} w_{x,x'} = \begin{cases} (1-\alpha) + n\alpha + nr\beta + r(\gamma - \beta), & \text{for } x \in \mathbb{D}_d, \\ (1-\alpha) + n\alpha + nr\beta, & \text{for } x \notin \mathbb{D}_d, \end{cases} \tag{34}$$

and correspondingly

$$\frac{w_{x,x'}}{w_x w_{x'}} = \begin{cases} \dfrac{1}{(1-\alpha) + n\alpha + nr\beta + r(\gamma - \beta)}, & \text{for } x = x', x \in \mathbb{D}_d, \\[2ex] \dfrac{1}{(1-\alpha) + n\alpha + nr\beta}, & \text{for } x = x', x \notin \mathbb{D}_d, \\[2ex] \dfrac{\alpha}{(1-\alpha) + n\alpha + nr\beta + r(\gamma - \beta)}, & \text{for } x \neq x', y(x) = y(x'), x, x' \in \mathbb{D}_d, \\[2ex] \dfrac{\alpha}{\sqrt{(1-\alpha) + n\alpha + nr\beta + r(\gamma - \beta)}\sqrt{(1-\alpha) + n\alpha + nr\beta}}, \\ \qquad \text{for } x \neq x', y(x) = y(x'), x \in \mathbb{D}_d \text{ or } x' \in \mathbb{D}_d, \\[2ex] \dfrac{\alpha}{(1-\alpha) + n\alpha + nr\beta}, & \text{for } x \neq x', y(x) = y(x'), x, x' \notin \mathbb{D}_d, \\[2ex] \dfrac{\gamma}{(1-\alpha) + n\alpha + nr\beta + r(\gamma - \beta)}, & \text{for } y(x) \neq y(x'), x, x' \in \mathbb{D}_d, \\[2ex] \dfrac{\beta}{\sqrt{(1-\alpha) + n\alpha + nr\beta + r(\gamma - \beta)}\sqrt{(1-\alpha) + n\alpha + nr\beta}}, \\ \qquad \text{for } y(x) \neq y(x'), x \in \mathbb{D}_d \text{ or } x' \in \mathbb{D}_d, \\[2ex] \dfrac{\beta}{(1-\alpha) + n\alpha + nr\beta}, & \text{for } y(x) \neq y(x'), x, x' \notin \mathbb{D}_d, \end{cases} \tag{35}$$

If we let

$$m_{x,x'} = \begin{cases} -\dfrac{r(\gamma-\beta)}{[(1-\alpha)+n\alpha+nr\beta+r(\gamma-\beta)]^2[(1-\alpha)+n\alpha+nr\beta]}, \\ \qquad\qquad\qquad\qquad\qquad\qquad\qquad \text{for } x=x', x\in\mathbb{D}_d, \\[4pt] -\dfrac{r(\gamma-\beta)}{[(1-\alpha)+n\alpha+nr\beta+r(\gamma-\beta)]^2[(1-\alpha)+n\alpha+nr\beta]}\alpha, \\ \qquad\qquad\qquad\qquad\qquad \text{for } x\neq x', y(x)=y(x'), x, x'\in\mathbb{D}_d, \\[4pt] -\dfrac{\frac{\sqrt{(1-\alpha)+n\alpha+nr\beta+r(\gamma-\beta)}}{\sqrt{(1-\alpha)+n\alpha+nr\beta}}-1}{[(1-\alpha)+n\alpha+nr\beta+r(\gamma-\beta)][(1-\alpha)+n\alpha+nr\beta]}\alpha, \\ \qquad\qquad\qquad\qquad \text{for } x\neq x', y(x)=y(x'), x\in\mathbb{D}_d \text{ or } x'\in\mathbb{D}_d, \\[4pt] \dfrac{[(1-\alpha)+n\alpha+(n-1)r\beta](\gamma-\beta)}{[(1-\alpha)+n\alpha+nr\beta+(\gamma-\beta)]^2[(1-\alpha)+n\alpha+nr\beta]}, \\ \qquad\qquad\qquad\qquad\qquad\qquad \text{for } y(x)\neq y(x'), x, x'\in\mathbb{D}_d, \\[4pt] -\dfrac{\frac{\sqrt{(1-\alpha)+n\alpha+nr\beta+r(\gamma-\beta)}}{\sqrt{(1-\alpha)+n\alpha+nr\beta}}-1}{[(1-\alpha)+n\alpha+nr\beta+r(\gamma-\beta)][(1-\alpha)+n\alpha+nr\beta]}\beta, \\ \qquad\qquad\qquad\qquad \text{for } y(x)\neq y(x'), x\in\mathbb{D}_d \text{ or } x'\in\mathbb{D}_d, \\[4pt] 0 \qquad\qquad\qquad\qquad\qquad\qquad\qquad\qquad\qquad\qquad \text{otherwise,} \end{cases} \tag{36}$$

then we have

$$\sqrt{w_x}\sqrt{w_{x'}}m_{x,x'}$$

$$= \begin{cases} -\dfrac{r(\gamma-\beta)}{[(1-\alpha)+n\alpha+nr\beta+r(\gamma-\beta)][(1-\alpha)+n\alpha+nr\beta]}, \\ \qquad\qquad\qquad\qquad\qquad\qquad\qquad \text{for } x=x', x\in\mathbb{D}_d, \\[4pt] -\dfrac{r(\gamma-\beta)}{[(1-\alpha)+n\alpha+nr\beta+r(\gamma-\beta)][(1-\alpha)+n\alpha+nr\beta]}\alpha, \\ \qquad\qquad\qquad\qquad\qquad \text{for } x\neq x', y(x)=y(x'), x, x'\in\mathbb{D}_d, \\[4pt] -\dfrac{\sqrt{(1-\alpha)+n\alpha+nr\beta+r(\gamma-\beta)}-\sqrt{(1-\alpha)+n\alpha+nr\beta}}{\sqrt{(1-\alpha)+n\alpha+nr\beta+r(\gamma-\beta)}[(1-\alpha)+n\alpha+nr\beta]}\alpha, \\ \qquad\qquad\qquad\qquad \text{for } x\neq x', y(x)=y(x'), x\in\mathbb{D}_d \text{ or } x'\in\mathbb{D}_d, \\[4pt] \dfrac{[(1-\alpha)+n\alpha+(n-1)r\beta](\gamma-\beta)}{[(1-\alpha)+n\alpha+nr\beta+(\gamma-\beta)][(1-\alpha)+n\alpha+nr\beta]}, \\ \qquad\qquad\qquad\qquad\qquad\qquad \text{for } y(x)\neq y(x'), x, x'\in\mathbb{D}_d, \\[4pt] -\dfrac{\sqrt{(1-\alpha)+n\alpha+nr\beta+r(\gamma-\beta)}-\sqrt{(1-\alpha)+n\alpha+nr\beta}}{\sqrt{(1-\alpha)+n\alpha+nr\beta+r(\gamma-\beta)}[(1-\alpha)+n\alpha+nr\beta]}\beta, \\ \qquad\qquad\qquad\qquad \text{for } y(x)\neq y(x'), x\in\mathbb{D}_d \text{ or } x'\in\mathbb{D}_d, \\[4pt] 0 \qquad\qquad\qquad\qquad\qquad\qquad\qquad\qquad\qquad\qquad \text{otherwise,} \end{cases} \tag{37}$$

and accordingly

$$\frac{w_{x,x'}}{w_x w_{x'}} - \sqrt{w_x}\sqrt{w_{x'}}m_{x,x'} = \begin{cases} \dfrac{1}{(1-\alpha)+n\alpha+nr\beta} & \text{for } x=x', \\[6pt] \dfrac{\alpha}{(1-\alpha)+n\alpha+nr\beta} & \text{for } x\neq x', y(x)=y(x'), \\[6pt] \dfrac{\beta}{(1-\alpha)+n\alpha+nr\beta} & \text{otherwise.} \end{cases} \tag{38}$$

In this case, $\bar{A} - \bar{M}$ is equivalent to the normalized similarity matrix of data without difficult examples. That is, we have

$$\mathcal{E}_{\mathrm{M}} = \mathcal{E}_{\mathrm{w.o.}}. \tag{39}$$

$\square$

The spectral contrastive loss with temperature $\boldsymbol{T} = (\tau_{x,x'})$ is defined as

$$\mathcal{L}_{\mathrm{T}}(\boldsymbol{x}; f) = -2\mathbb{E}_{x,x^+} \frac{f(x)^\top f(x^+)}{\tau_{x,x^+}} + \mathbb{E}_{x,x'}\left[\frac{f(x)^\top f(x')}{\tau_{x,x'}}\right]^2. \tag{40}$$

*Proof of Theorem 4.4.*

$$\mathcal{L}_{\mathrm{T}} = \mathbb{E}_{x,x^+} f(x)^\top f(x^+)/\tau_{x,x^+} + \mathbb{E}_{x,x'}\left[f(x)^\top f(x')/\tau_{x,x'}\right]^2$$

$$= -2\sum_{x,x^+} w_{x,x'} f(x)^\top f(x^+)/\tau_{x,x^+} + \sum_{x,x'} w_x w_{x'}\left[f(x)^\top f(x')/\tau_{x,x'}\right]^2$$

$$= \sum_{x,x'}\left\{ -2 w_{x,x'}/\tau_{x,x'} f(x)^\top f(x^+) + w_x w_{x'}/\tau_{x,x'}^2 \left[f(x)^\top f(x')/\tau_{x,x'}\right]^2 \right\}$$

$$= \sum_{x,x'}\left\{ -2\frac{1}{\tau_{x,x'}}\frac{w_{x,x'}}{\sqrt{w_x}\sqrt{w_{x'}}}[\sqrt{w_x}f(x)]^\top[\sqrt{w_{x'}}f(x')] + \frac{1}{\tau_{x,x'}^2}\left[[\sqrt{w_x}f(x)]^\top[\sqrt{w_{x'}}f(x')]\right]^2 \right\}$$

$$= \sum_{x,x'}\frac{1}{\tau_{x,x'}^2}\left\{ \left[[\sqrt{w_x}f(x)]^\top[\sqrt{w_{x'}}f(x')]\right]^2 - 2\frac{\tau_{x,x'} w_{x,x'}}{\sqrt{w_x}\sqrt{w_{x'}}}[\sqrt{w_x}f(x)]^\top[\sqrt{w_{x'}}f(x')] \right.$$

$$\left. + \frac{\tau_{x,x'}^2 w_{x,x'}^2}{w_x w_{x'}} - \frac{\tau_{x,x'}^2 w_{x,x'}^2}{w_x w_{x'}} \right\}$$

$$= \sum_{x,x'}\frac{1}{\tau_{x,x'}^2}\left[\tau_{x,x'}\frac{w_{x,x'}}{\sqrt{w_x}\sqrt{w_{x'}}} - [\sqrt{w_x}f(x)]^\top[\sqrt{w_{x'}}f(x')]\right]^2 - \frac{1}{\tau_{x,x'}^2}\sum_{x,x'}\frac{\tau_{x,x'}^2 w_{x,x'}^2}{w_x w_{x'}}$$

$$:= \|\boldsymbol{T} \odot \bar{\boldsymbol{A}} - FF^\top\|_{wF}^2 - \frac{1}{\tau_{x,x'}^2}\sum_{x,x'}\frac{\tau_{x,x'}^2 w_{x,x'}^2}{w_x w_{x'}}, \tag{41}$$

where we denote $\boldsymbol{T} := (\tau_{x,x'})_{x,x'\in\{x_i\}_{i=1}^{n(r+1)}}$, $\bar{\boldsymbol{A}} := \boldsymbol{D}^{-1/2}\boldsymbol{A}\boldsymbol{D}^{-1/2}$, $\boldsymbol{A} := (w_{x,x'})_{x,x'\in\{x_i\}_{i=1}^{n(r+1)}}$, $\boldsymbol{D} := \mathrm{diag}(w_1,\ldots,w_{n(r+1)})$, $F = (\sqrt{w_x}f(x))_{x\in\{x_i\}_{i=1}^{n(r+1)}}$, $\boldsymbol{T} \odot \bar{\boldsymbol{A}}$ as the element-wise product of matrices $\boldsymbol{T}$ and $\bar{\boldsymbol{A}}$, and $\|\cdot\|_{wF}$ as the weighted Frobenius norm where $\|\boldsymbol{B}\|_{wF}^2 := \sum_{x,x'}\frac{1}{\tau_{x,x'}^2}b_{x,x'}^2$ for arbitrary matrix $\boldsymbol{B} = (b_{x,x'}) \in \mathbb{R}^{n(r+1)\times n(r+1)}$.

Note that given the adjacency matrix of the similarity graph $\boldsymbol{A}$ and the temperature matrix $\boldsymbol{T}$, the second term in equation 41 is a constant. Therefore, minimizing the temperature scaling loss $\mathcal{L}_{\mathrm{T}}$ over $f(x)$ is equivalent to minimizing the matrix factorization loss $\mathcal{L}_{\mathrm{mf-T}} := \|\boldsymbol{T} \odot \bar{\boldsymbol{A}} - FF^\top\|_{wF}^2$ over $F$. $\qquad\square$

Before we proceed to the proof of Theorem 4.5, we first extend Theorem B.3 in HaoChen et al. (2021) to the temperature scaling loss by deriving the matrix factorization error bound under the weighted Frobenius norm.

**Lemma B.1.** *Let $f_{\mathrm{pop}}^* \in \arg\min_{f:\mathcal{X}\to\mathbb{R}^k} \mathcal{L}_{\mathrm{T}}(f)$ be a minimizer of the population temperature-scaling loss $\mathcal{L}_{\mathrm{T}}(f)$. Then for any labeling function $\hat{y} : \mathcal{X} \to [r]$, there exists a linear probe $B^* \in \mathbb{R}^{r\times k}$ with norm $\|B^*\|_F \leq 1/(1-\lambda_k)$ such that*

$$\mathbb{E}_{\bar{x}\sim\mathcal{P}_{\bar{X}},x\sim\mathcal{A}(\cdot|\bar{x})}\left[\|\vec{y} - B^* f_{\mathrm{pop}}^*(x)\|_2^2\right] \leq \frac{\tilde{\phi}^{\hat{y}}}{1-\lambda_{k+1}} + 4\Delta(y,\hat{y}), \tag{42}$$

*where $\vec{y}(\bar{x})$ is the one-hot embedding of $y(\bar{x})$, and*

$$\tilde{\phi}^{\hat{y}} = \sum_{x,x'\sim\mathcal{X}}\frac{w_{x,x'}}{\tau_{x,x'}^2}\mathbf{1}[\hat{y}(x) \neq \hat{y}(x')]. \tag{43}$$

*Furthermore, the error can be bounded by*

$$\mathcal{E}_{\mathrm{T}} = \Pr_{\bar{x}\sim\mathcal{P}_{\bar{X}},x\sim\mathcal{A}(\cdot|\bar{x})}\left(g_{f_{\mathrm{pop}^*},B^*}(x) \neq y(\bar{x})\right) \leq \frac{2\tilde{\phi}^{\hat{y}}}{1-\lambda_{k+1}} + 8\Delta(y,\hat{y}). \tag{44}$$

We also need the following two supporting lemmas to prove Lemma B.1.

**Lemma B.2.** *Let $L$ be the normalized Laplacian matrix of some graph $G$, $v_i$ be the $i$-th smallest unit-norm eigenvector of $\boldsymbol{L}$ with eigenvalue $1 - \lambda_i$, and $\tilde{R}(u) := \frac{\tilde{u}^\top \boldsymbol{L}\tilde{u}}{u^\top u}$ for a vector $u \in \mathbb{R}^N$, where $\tilde{u} = (u_i/\tau_i)_{i=1}^N$. Then for any $k \in \mathbb{Z}^+$ such that $k < N$ and $1 - \lambda_{k+1} > 0$, there exists a vector $b \in \mathbb{R}^k$ with norm $\|b\|_2 \leq \|u\|_2$ such that*

$$\left\| u - \sum_{i=1}^k b_i v_i \right\|_w^2 \leq \frac{\tilde{R}(u)}{1 - \lambda_{k+1}} \|u\|_2^2, \tag{45}$$

*where $\|\cdot\|$ denotes the weighted $l^2$-norm with weights $\tau^{-2} = (1/\tau_i^2)_{i=1}^N$.*

*Proof of Lemma B.2.* We can decompose the vector $u$ in the eigenvector basis as

$$u = \sum_{i=1}^N \zeta_i v_i. \tag{46}$$

Let $b \in \mathbb{R}^k$ be the vector such that $b_i = \zeta_i$. Then we have $\|b\|_2^2 \leq \|u\|_2^2$ and

$$\begin{aligned}
\left\| u - \sum_{i=1}^k b_i v_i \right\|_w^2 &= \| \sum_{i=k+1}^N \zeta_i v_i \|_w^2 \\
&= \sum_{i=k+1}^N \zeta_i^2/\tau_i^2 \\
&\leq \frac{1}{1 - \lambda_{k+1}} \sum_{i=k+1}^N (1 - \lambda_i)\zeta_i^2/\tau_i^2 \\
&= \frac{1}{1 - \lambda_{k+1}} \sum_{i=k+1}^N \zeta_i^2/\tau_i^2 v_i^\top (1 - \lambda_i)v_i \\
&= \frac{1}{1 - \lambda_{k+1}} \sum_{i=k+1}^N \zeta_i^2/\tau_i^2 v_i^\top \boldsymbol{L} v_i \\
&= \frac{1}{1 - \lambda_{k+1}} \sum_{i=k+1}^N (\zeta_i/\tau_i \cdot v_i)^\top \boldsymbol{L}(\zeta_i/\tau_i \cdot v_i). \tag{47}
\end{aligned}$$

Denote $\tilde{u} = \sum_{i=1} \zeta_i/\tau_i \cdot v_i$ and $\tilde{R}(u) := \frac{\tilde{u}^\top \boldsymbol{L}\tilde{u}}{u^\top u}$. Then we have

$$\left\| u - \sum_{i=1}^k b_i v_i \right\|_w^2 \leq \frac{\tilde{R}(u)}{1 - \lambda_{k+1}} \|u\|_2^2. \tag{48}$$

$\square$

**Lemma B.3.** *In the setting of Lemma B.2, let $\hat{y}$ be an extended labeling function. Fix $i \in [r]$. Define function $u_i^{\hat{y}}(x) := \sqrt{w_x} \cdot \mathbf{1}[\hat{y}(x) = i]$ and $u_i^{\hat{y}}$ is the corresponding vector in $\mathbb{R}^N$. Also define the following quantity*

$$\tilde{\phi}_i^{\hat{y}} := \frac{\sum_{x,x' \in \mathcal{X}} w_{x,x'}/\tau_{x,x'}^2 \cdot \mathbf{1}[(\hat{y}(x) = i \wedge \hat{y}(x') \neq i) \text{ or } (\hat{y}(x) \neq i \wedge \hat{y}(x') = i)]}{\sum_{x \in \mathcal{X}} w_x \cdot \mathbf{1}[\hat{y}(x) = i]}. \tag{49}$$

*Then we have*

$$\tilde{R}(u_i^{\hat{y}}) = \frac{1}{2}\tilde{\phi}_i^{\hat{y}}. \tag{50}$$

*Proof of Lemma B.3.* Let $f$ be any function $\mathcal{X} \to \mathbb{R}$, define function $u(x) := \sqrt{w_x} \cdot f(x)$. Let $u \in \mathbb{R}^N$ be the vector corresponding to $u$. Then by definition of Laplacian matrix, we have

$$\tilde{u}^\top \boldsymbol{L}\tilde{u} = \|\tilde{u}\|_2^2 - \tilde{u}\boldsymbol{D}^{-1/2}\boldsymbol{A}\boldsymbol{D}^{-1/2}\tilde{u}$$

$$= \sum_{x \in \mathcal{X}} w_x / \tau_x^2 f(x)^2 - \sum_{x,x' \in \mathcal{X}} w_{x,x'} / \tau_{x,x'}^2 f(x) f(x')$$

$$= \frac{1}{2} \sum_{x,x' \in \mathcal{X}} w_{x,x'} / \tau_{x,x'}^2 [f(x) - f(x')]^2. \tag{51}$$

Therefore we have

$$\tilde{R}(u_i^{\hat{y}}) = \frac{1}{2} \frac{\sum_{x,x' \in \mathcal{X}} w_{x,x'} / \tau_{x,x'}^2 [f(x) - f(x')]^2}{\sum_{x \in \mathcal{X}} w_x f(x)^2}. \tag{52}$$

Setting $f(x) = \mathbf{1}[\hat{y}(x) = i]$ finishes the proof. $\qquad \square$

*Proof of Lemma B.1.* Let $F_{\text{sc}} = [v_1, v_2, \ldots, v_k]$ be the matrix that contains the smallest $k$ eigenvectors of $\boldsymbol{L} = \boldsymbol{I} - \bar{\boldsymbol{A}}$ as columns, and $f_{\text{sc}}$ is the corresponding feature extractor. By Lemma B.2, there exists a vector $b_i \in \mathbb{R}^k$ with norm bound $\|b_i\|_2 \leq \|u_i^{\hat{y}}\|_2$ such that

$$\|u_i^{\hat{y}} - F_{\text{sc}} b_i\|_w^2 \leq \frac{\tilde{R}(u_i^{\hat{y}})}{1 - \lambda_{k+1}} \|u_i^{\hat{y}}\|_2^2. \tag{53}$$

Combined with Lemma B.3, we have

$$\|u_i^{\hat{y}} - F_{\text{sc}} b_i\|_w^2 \leq \frac{\tilde{\phi}_i^{\hat{y}}}{2(1 - \lambda_{k+1})} \cdot \sum_{x \in \mathcal{X} \cdot \mathbf{1}[\hat{y}(x) = i]}$$

$$= \frac{1}{2(1 - \lambda_{k+1})} \sum_{x,x' \in \mathcal{X}} w_{x,x'} / \tau_{x,x'}^2 \cdot \mathbf{1}[(\hat{y}(x) = i \wedge \hat{y}(x') \neq i) \text{ or } (\hat{y}(x) \neq i \wedge \hat{y}(x') = i)]. \tag{54}$$

Let matrix $U := (u_i^{\hat{y}})_{i=1}^k$, and let $u : \mathcal{X} \to \mathbb{R}^k$ be the corresponding feature extractor. Define matrix $B \in \mathbb{R}^{N \times k}$ such that $B^\top = (b_1, \ldots, b_k)$. Summing equation 54 over all $i \in [k]$ and by definition of $\tilde{\phi}^{\hat{y}}$ we have

$$\|U - F_{\text{sc}} B^\top\|_{wF}^2 \leq \frac{1}{2(1 - \lambda_{k+1})} \sum_{x,x' \in \mathcal{X}} w_{x,x'} / \tau_{x,x'}^2 \cdot \mathbf{1}[\hat{y}(x) \neq \hat{y}(x')] = \frac{\tilde{\phi}^{\hat{y}}}{2(1 - \lambda_{k+1})}. \tag{55}$$

By Theorem 4.4, for a feature extractor $f_{\text{pop}}^*$ that minimizes the temperature scaling loss $\mathcal{L}_{\tilde{T}}$, the function $f_{\text{mf}}^*(x) := \sqrt{w_x} \cdot f_{\text{pop}}^*$ is a minimizer of the matrix factorization loss $\mathcal{L}_{\text{mf}-\text{T}}$. Then we have

$$\mathbb{E}_{\bar{x} \sim \mathcal{P}_{\bar{X}}, x \sim \mathcal{A}(\cdot | \bar{x})} \|\vec{y}(x) - B^* f_{\text{pop}}^*(x)\|_2^2$$

$$\leq 2\mathbb{E}_{\bar{x} \sim \mathcal{P}_{\bar{X}}, x \sim \mathcal{A}(\cdot | \bar{x})} \|\vec{\hat{y}}(x) - B^* f_{\text{pop}}^*(x)\|_2^2 + 2\mathbb{E}_{\bar{x} \sim \mathcal{P}_{\bar{X}}, x \sim \mathcal{A}(\cdot | \bar{x})} \|\vec{\hat{y}}(x) - \vec{y}(x)\|_2^2$$

$$= 2 \sum_{x \in \mathcal{X}} w_x \cdot \|\vec{\hat{y}}(x) - B^* f_{\text{pop}}^*(x)\|_2^2 + 4\Delta(y, \hat{y})$$

$$= 2\|U - F_{\text{sc}} B^\top\|_{wF}^2 + 4\Delta(y, \hat{y})$$

$$\leq \frac{\tilde{\phi}^{\hat{y}}}{1 - \lambda_{k+1}} + 4\Delta(y, \hat{y}). \tag{56}$$

$$\square$$

Then we move on to the formal proof of Theorem 4.5.

*Proof of Theorem 4.5.* According to equation 35 the proof of Theorem 4.3, if we let

$$\tau_{x,x'} = \begin{cases} \dfrac{(1-\alpha) + n\alpha + nr\beta + r(\gamma - \beta)}{(1-\alpha) + n\alpha + nr\beta}, & \text{for } y(x) = y(x'), x, x' \in \mathbb{D}_d, \\[3mm] \dfrac{\sqrt{(1-\alpha) + n\alpha + nr\beta + r(\gamma - \beta)}}{\sqrt{(1-\alpha) + n\alpha + nr\beta}}, & \text{for } x \in \mathbb{D}_d \text{ or } x' \in \mathbb{D}_d, \\[3mm] \dfrac{[(1-\alpha) + n\alpha + nr\beta + r(\gamma - \beta)]\beta}{[(1-\alpha) + n\alpha + nr\beta]\gamma}, & \text{for } y(x) \neq y(x'), x, x' \in \mathbb{D}_d, \\[3mm] 1, & \text{otherwise,} \end{cases} \tag{57}$$

then we have

$$\tau_{x,x'} \cdot \frac{w_{x,x'}}{w_x w_{x'}} = \begin{cases} \dfrac{1}{(1-\alpha)+n\alpha+nr\beta} & \text{for } x = x', \\[2ex] \dfrac{\alpha}{(1-\alpha)+n\alpha+nr\beta} & \text{for } x \neq x', y(x) = y(x'), \\[2ex] \dfrac{\beta}{(1-\alpha)+n\alpha+nr\beta} & \text{otherwise.} \end{cases} \tag{58}$$

In this case, $\boldsymbol{T} \odot \bar{\boldsymbol{A}}$ is equivalent to the normalized similarity matrix of data without difficult examples.

By Lemma B.1, we have

$$\mathcal{E}_{\mathrm{T}} \leq \frac{2\tilde{\phi}^{\hat{y}}}{1-\lambda_{k+1}} + 8\Delta(y, \hat{y}). \tag{59}$$

By Assumption 3.1, we have $\Delta(y, \hat{y}) \leq \delta$. Besides, since $\tau_{x,x'} \leq 1$ for $y(x) \neq y(x')$, $x, x' \in \mathbb{D}_{\mathrm{d}}$, and otherwise $\tau_{x,x'} \geq 1$, we have

$$\tilde{\phi}^{\hat{y}} = \sum_{x,x' \in \mathcal{X}} w_{x,x'}/\tau_{x,x'}^2 \mathbf{1}[\hat{y}(x) \neq \hat{y}(x')]$$

$$\leq \sum_{x,x' \in \mathcal{X}\backslash\{x,x':x,x'\in\mathbb{D}_{\mathrm{d}}\}} w_{x,x'}\mathbf{1}[\hat{y}(x) \neq \hat{y}(x')] + \sum_{y(x)\neq y(x'),x,x'\in\mathbb{D}_{\mathrm{d}}} (\gamma/\beta)^2 w_{x,x'}\mathbf{1}[\hat{y}(x) \neq \hat{y}(x')]$$

$$= \sum_{x,x' \in \mathcal{X}\backslash\{x,x':x,x'\in\mathbb{D}_{\mathrm{d}}\}} \mathbb{E}_{\bar{x}\sim\mathcal{P}_{\bar{\mathcal{X}}}}[\mathcal{A}(x|\bar{x})\mathcal{A}(x'|\bar{x}) \cdot \mathbf{1}[\hat{y}(x) \neq \hat{y}(x')]]$$

$$+ (\gamma/\beta)^2 \sum_{x,x' \in \mathbb{D}_{\mathrm{d}}} \mathbb{E}_{\bar{x}\sim\mathcal{P}_{\bar{\mathcal{X}}}}[\mathcal{A}(x|\bar{x})\mathcal{A}(x'|\bar{x}) \cdot \mathbf{1}[\hat{y}(x) \neq \hat{y}(x')]]$$

$$\leq \sum_{x,x' \in \mathcal{X}\backslash\{x,x':x,x'\in\mathbb{D}_{\mathrm{d}}\}} \mathbb{E}_{\bar{x}\sim\mathcal{P}_{\bar{\mathcal{X}}}}[\mathcal{A}(x|\bar{x})\mathcal{A}(x'|\bar{x}) \cdot (\mathbf{1}[\hat{y}(x) \neq \hat{y}(\bar{x})] + \mathbf{1}[\hat{y}(x') \neq \hat{y}(\bar{x})])]$$

$$+ (\gamma/\beta)^2 \sum_{x,x' \in \mathbb{D}_{\mathrm{d}}} \mathbb{E}_{\bar{x}\sim\mathcal{P}_{\bar{\mathcal{X}}}}[\mathcal{A}(x|\bar{x})\mathcal{A}(x'|\bar{x}) \cdot (\mathbf{1}[\hat{y}(x) \neq \hat{y}(\bar{x})] + \mathbf{1}[\hat{y}(x') \neq \hat{y}(\bar{x})])]$$

$$= 2[1 - (n_d/n)^2]\mathbb{E}_{\bar{x}\sim\mathcal{P}_{\bar{\mathcal{X}}}}[\mathcal{A}(x|\bar{x}) \cdot \mathbf{1}[\hat{y}(x) \neq \hat{y}(\bar{x})]]$$

$$+ 2(\gamma/\beta)^2 (n_d/n)^2 \mathbb{E}_{\bar{x}\sim\mathcal{P}_{\bar{\mathcal{X}}}}[\mathcal{A}(x|\bar{x}) \cdot \mathbf{1}[\hat{y}(x) \neq \hat{y}(\bar{x})]]$$

$$= 2[1 - (n_d/n)^2 + (\gamma/\beta)^2 (n_d/n)^2]\delta. \tag{60}$$

Therefore we have

$$\mathcal{E}_{\mathrm{T}} \leq \frac{2\tilde{\phi}^{\hat{y}}}{1-\lambda_{k+1}} + 8\Delta(y, \hat{y}) \leq [1 - (n_d/n)^2 + (\gamma/\beta)^2 (n_d/n)^2] \cdot \frac{4\delta}{1 - \frac{1-\alpha}{(1-\alpha)+n\alpha+nr\beta}} + 8\delta. \tag{61}$$

$$\square$$

## B.3 RELAXATION ON THE IDEAL ADJACENCY MATRIX

To enhance the connection of the theoretical modeling of difficult examples (Section 3.2) to real-world scenarios, we hereby discuss a possible relaxation on the ideal adjacency matrix of the similarity graph.

The adjacency matrix could be relaxed by adding random terms to the similarity values. Specifically, we replace $\boldsymbol{A}$ with $\tilde{\boldsymbol{A}} = (\tilde{a}_{ij})$, where $\tilde{a}_{ii} = 1$, and $\tilde{a}_{ij} = \tilde{a}_{ij} + \epsilon \cdot \varepsilon_{ij}$ for $i \neq j$, $a_{ij}$ takes values in $\{\alpha, \beta, \gamma\}$, $\varepsilon_{ij} = \varepsilon_{ji}$ are i.i.d. random variables with mean 0 and variance 1, $\epsilon > 0$ is a small constant. Then $\tilde{\boldsymbol{A}}$ can be decomposed into

$$\tilde{\boldsymbol{A}} = \boldsymbol{A} + \epsilon \cdot \boldsymbol{W} - \epsilon \cdot \mathrm{diag}(\varepsilon_{ii}), \tag{62}$$

where $W$ turns out to be a real Wigner matrix. Note that as $\mathbb{E}\varepsilon_{ij} = 0$, the normalization matrix $\tilde{D} \rightarrow \mathbb{E}\tilde{D} = D$, as $n(r+1) \rightarrow \infty$, and therefore we have $\bar{\tilde{A}} = \tilde{D}^{-1/2}\tilde{A}\tilde{D}^{-1/2} \approx D^{-1/2}\tilde{A}D^{-1/2}$.

For mathematical convenience, in the following analysis, we instead perform the relaxation on the normalized adjacency matrix $\bar{A}$, and investigate

$$\bar{\tilde{A}} = \bar{A} + \epsilon' \cdot W' - \epsilon' \cdot \text{diag}(\varepsilon_{ii}), \tag{63}$$

where $\epsilon > 0$ and $W$ is a Wigner's matrix.

**Theorem B.4** (Generalized version of Theorem 3.3). *Denote $\mathcal{E}_{\text{w.o.}}$ as the linear probing error of a contrastive learning model trained on a dataset without difficult examples. Under the generalized assumption that $A' = A + \epsilon W$, where $W$ is a Wigner matrix with $\varepsilon_{ii}$ following the Dirac distribution, then if $n(r+1)$ is large enough, we have*

$$\mathcal{E}_{\text{w.o.}} \leq \frac{4\delta}{1 - \frac{1-\alpha}{(1-\alpha)+n\alpha+nr\beta} - \frac{1}{(1-\alpha)+n(\alpha+r\beta)}x_0 \cdot \epsilon} + 8\delta,$$

*where $x_0 \in (0,2)$ is the unique solution to the following Kepler's equation*

$$\frac{1}{2}x_0\sqrt{4-x_0^2} + 2\arg\sin(x_0/2) = \left[1 - \frac{2}{r+1}\frac{n_d}{n}\right]\pi.$$

**Theorem B.5** (Generalized version of Theorem 3.4). *Denote $\mathcal{E}_{\text{w.d.}}$ as the linear probing error of a contrastive learning model trained on a dataset with $n_d$ difficult examples per class. Under the generalized assumption that $A' = A + \epsilon W$, where $W$ is a Wigner matrix with $\varepsilon_{ii}$ following the Dirac distribution, if $n(r+1)$ is large enough and $r+1 \leq k < n_d+r+1$, we have*

$$\mathcal{E}_{\text{w.d.}} \leq \frac{4\delta}{1 - \frac{(1-\alpha)+r(\gamma-\beta)}{(1-\alpha)+n\alpha+nr\beta+r(\gamma-\beta)} - \frac{1}{(1-\alpha)+n(\alpha+r\beta)}x_0 \cdot \epsilon} + 8\delta.$$

**Remark 1.** (Value of $x_0$) We derive the value of $x_0$ through numerical methods for multiple datasets, by using the empirical values of $\alpha$, $\beta$, and $\gamma$ calculated on the proxy augmentation graph. We have $x_0 = 1.894$ for CIFAR-10, $x_0 = 1.976$ for CIFAR-100, and $x_0 = 1.995$ for Imagenet-1k. Intuitively, according to Wigner's Semicircle Law, because $n_d \ll n(r+1)$, the value of $x_0$ is near 2.

**Remark 2.** (Range of $\epsilon$) The bounds are valid if $\frac{1-\alpha}{(1-\alpha)+n\alpha+nr\beta} + \frac{1}{(1-\alpha)+n(\alpha+r\beta)}x_0 \cdot \epsilon < 1$ and $\frac{(1-\alpha)+r(\gamma-\beta)}{(1-\alpha)+n\alpha+nr\beta+r(\gamma-\beta)} + \frac{1}{(1-\alpha)+n(\alpha+r\beta)}x_0 \cdot \epsilon < 1$. As $\frac{(1-\alpha)+r(\gamma-\beta)}{(1-\alpha)+n\alpha+nr\beta+r(\gamma-\beta)} > \frac{1-\alpha}{(1-\alpha)+n\alpha+nr\beta}$, we require $\epsilon < \epsilon_{\text{bound}} = \frac{1-\frac{(1-\alpha)+r(\gamma-\beta)}{(1-\alpha)+n\alpha+nr\beta+r(\gamma-\beta)}}{\frac{1}{(1-\alpha)+n(\alpha+r\beta)}x_0}$.

**Remark 3.** (Existing conclusions still hold) Note that the effect of the random similarity $\epsilon \cdot \varepsilon_{ij}$ is to add an additional term to the upper bound of the eigenvalue, and the effect is the same with and without the existence of difficult examples. When $\epsilon = 0$, Theorems B.4 and B.5 degenerates to Theorems 3.3 and 3.4. . Moreover, as Corollary 4.1 can be directly derived by Theorem 3.3, the generalized version becomes $\mathcal{E}_{\text{R}} \leq \frac{4\delta}{1-\frac{1-\alpha}{(1-\alpha)+(n-n_d)\alpha+(n-n_d)r\beta} - \frac{1}{(1-\alpha)+(n-n_d)(\alpha+r\beta)}x_0 \cdot \epsilon} + 8\delta$. Similarly, as the bounds in Theorems 4.3 and 4.5 are based on a modified similarity matrix, we have $\mathcal{E}_{\text{M}} = \mathcal{E}_{\text{w.o.}}$ (Theorem 4.3) and $\mathcal{E}_{\text{T}} \leq \frac{4[1-(n_d/n)^2+(\gamma/\beta)^2(n_d/n)^2]\delta}{1-\frac{1-\alpha}{(1-\alpha)+n\alpha+nr\beta} - \frac{1}{(1-\alpha)+n(\alpha+r\beta)}x_0 \cdot \epsilon} + 8\delta$ (Theorem 4.5), where the theoretical insights of these two theorems remain unchanged. That is, even under the generalized assumptions, we still have the conclusion that sample removal, margin tuning, and temperature scaling improve the error bound under the existence of difficult examples.

*Proof.* Because $\mathbb{E}W = \mathbf{0}$, when $n(r+1)$ is large enough, after normalization, we have $\bar{A}' = \bar{A} + \frac{1}{(1-\alpha)+n(\alpha+r\beta)} \cdot \epsilon W$. By Equation 13 in Fulton (2000), when $r+1 \leq k < n_d+r+1$, we have the $k+1$-th largest eigenvalue of $\bar{A}'$ satisfying

$$\lambda'_{k+1} \leq \min_{i+j=k+2} \lambda_i + \frac{1}{(1-\alpha)+n(\alpha+r\beta)} \cdot \epsilon\nu_j \leq \lambda_{r+2} + \frac{1}{(1-\alpha)+n(\alpha+r\beta)} \cdot \epsilon\nu_{n_d},$$

where $\lambda_i$ is the $i$-th largest eigenvalue of $\bar{A}$ and $\nu_j$ is the $j$-th largest eigenvalue of $W$.

On the one hand, according to the proofs of Theorems 3.3 and 3.4, we have $\lambda_{r+2} = \frac{1-\alpha}{(1-\alpha)+n(\alpha+r\beta)}$ (Theorem 3.3) and $\lambda_{r+2} \leq \frac{(1-\alpha)+r(\gamma-\beta)}{(1-\alpha)+n(\alpha+r\beta)+r(\gamma-\beta)}$ (Theorem 3.3).

On the other hand, Because $\boldsymbol{W}$ is a Wigner matrix, we have its empirical spectral measure $\nu = \frac{1}{n(r+1)} \sum_{i=1}^{n(r+1)} \delta_{\nu_i}$ converging weakly almost surely to the quarter-circle distribution on $[0,2]$, with density $f(\nu) = \frac{1}{2\pi}\sqrt{4-x^2}\mathbf{1}[|x| \leq 2]$. When $j \leq n(r+1)/2$ and $n(r+1)$ large enough, by symmetry of $f(\nu)$, we have

$$\frac{1}{2}\Big[1 - \frac{2j}{n(r+1)}\Big] = \int_{x=0}^{\nu_j} f(\nu), d\nu = \frac{1}{2\pi}\Big[\frac{1}{2}\nu_j\sqrt{4-\nu_j^2} + 2\arg\sin(\nu_j/2)\Big]. \qquad (64)$$

Then combine the above calculations, we have $\lambda'_{k+1} \leq \frac{1-\alpha}{(1-\alpha)+n(\alpha+r\beta)} + \nu_j$ for the generalized Theorem 3.1 and $\lambda'_{k+1} \leq \frac{(1-\alpha)+r(\gamma-\beta)}{(1-\alpha)+n(\alpha+r\beta)+r(\gamma-\beta)} + \frac{1}{(1-\alpha)+n(\alpha+r\beta)} \cdot \epsilon\nu_j$, where $\nu_j$ is the solution to equation 64. Then we complete the proof by deriving the error bounds using the upper bounds of $\lambda'_{k+1}$. $\qquad\square$

## B.4 Discussions on the Tightness of Bounds

As a possible extension, we discuss that a potential lower bound related to the $\lambda_k$, which according to similar proofs of Theorem 3.4, also increases with $\gamma - \beta$ increasing. Besides, in Appendix A.5, we empirically verify that the trending of the downstream classification error, which also verify the reliability of our upper bounds. Please see the following for more details.

To derive the potential lower bound, we are mainly required to modify Theorem B.6 in HaoChen et al. (2021) to an alternative "lower bound" form (because our analysis is based on Theorem B.3 in HaoChen et al. (2021) and a key step to its proof is Theorem B.6). Specifically, with the notations of Theorem B.6 (while replacing $\lambda_i$ with $1 - \lambda_i$ to be consistent with our submission), we have

$$\|u - \sum_{i=1}^{k} b_i v_i\|_2^2 = \sum_{i=k+1}^{N} \zeta_i^2 \geq \frac{1}{(1-\lambda_N)}\Big[\sum_{i=1}^{N}(1-\lambda_i)\zeta_i^2 - \sum_{i=1}^{k}(1-\lambda_i)\zeta_i^2\Big]$$

$$\geq \frac{1}{(1-\lambda_N)}[R\|u\|_2^2 - (1-\lambda_k)\sum_{i=1}^{k}\|b_i\|^2]. \qquad (65)$$

According to Fulton (2000), we have $\lambda_k \geq \max_{i+j=n+k}\lambda_{1,i} + \lambda_{2,j}$, that is, we have $\lambda_N \geq [(1-\alpha) - r(\gamma-\beta)]/c_1$ and $\lambda_k \geq [(1-\alpha) + r(\gamma-\beta)]/c_1$ for $k \leq n_d$. Additionally, by Claim B.7 and Lemma B.8 in HaoChen et al. (2021) and some additional assumption that $\sum_{i=1}^{k}\|b_i\|^2 \leq C$, where $C > 0$ is a constant, we obtain the lower bound as

$$\mathcal{E}_{\text{w.d.}} \geq \frac{1}{1 - \frac{(1-\alpha)-r(\gamma-\beta)}{c_1}}\Big[\frac{1}{2}\phi^{\hat{y}} - 1 + \frac{(1-\alpha)+r(\gamma-\beta)}{c_1}C\Big]. \qquad (66)$$

Note that $\frac{1}{1-\frac{(1-\alpha)-r(\gamma-\beta)}{c_1}}$ decreases and $\frac{1}{1-\frac{(1-\alpha)-r(\gamma-\beta)}{c_1}} \cdot \frac{(1-\alpha)+r(\gamma-\beta)}{c_1}$ increases as $\gamma - \beta$ increases. Then when the labeling error $\phi^{\hat{y}}$ is relatively small, the lower bound gets larger with $\gamma - \beta$ increasing, which coincides with the insight of Theorem 3.4.

In addition, in Table 10, we empirically show the trending of our derived bounds, which also to some extend indicates the tightness of bounds. Specifically, on a Mixed CIFAR synthetic dataset, we show that as the difficult examples increases, the classification error and the bound share the same variation trend, thus validating Theorem 3.4 that a larger $\gamma - \beta$ results in worse error.

## C Usage of LLM

We commit to using LLMs for text polishing based on prompts. All polished text are double-checked by authors to ensure accuracy, avoid over-claims, and prevent confusion.

