# OpenReview forum: "Difficult Examples Hurt Unsupervised Contrastive Learning: A Theoretical Perspective"
_ICLR.cc/2026/Conference — ICLR 2026 Oral_

### Official Review · Reviewer_Vvxq · 2025-10-30

**Soundness:** 3
**Presentation:** 2
**Contribution:** 3
**Rating:** 6
**Confidence:** 4

**Summary:**

This paper builds upon the theoretical framework for spectral contrastive learning on population data by HaoChen et al. (2021) by defining a similarity graph. The authors utilize properties of the aforementioned framework to demonstrate in theory that difficult examples (samples near a class decision boundary in a supervised setting) hurt the generalization performance of unsupervised contrastive learning. The authors validate their theoretical work by proposing and validating the performance of mechanisms for detecting and dealing with difficult examples.

**Strengths:**

The mathematical framework utilized is a simplified (neural collapsed) version of a proven one used in HaoChen et al., 2021. The mathematical derivations in this framework are sound. The theoretical results here can be important for choices of contrastive learning samples.

**Weaknesses:**

The experimental section is only partially sound, with a limited number of experiments and hyperparameter tuning.

The presentation needs to be significantly improved. There are typographical errors throughout the paper. Figures 1, 2, 4 contain text that is too small to read. Certain ideas are not explained clearly enough.

The theoretical framework of this paper is based heavily on (HaoChen et al., 2021), which argues extensively about the similarities and differences between the empirical and augmented sets. None of this discussion is reflected here.


The results of experimentation in Section 5 are only averaged out of 3 runs. This makes most results (almost all that are not on TinyImagenet) close enough to be within margin of error. This is exacerbated by the fact that, as shown in Appendix A.3-A.4, only limited hyperparameter tuning is done. Moreover, TinyImagenet is the only dataset that uses a ResNet-50 model instead of ResNet-18 - a natural question arises: could the performance increase be related to the complexity of the model?

**Questions:**

What exactly does the mixing process itself look like in the mixing image experiment (Section 2)?

Section 3.2 Figure 3(d) shows the adjacency matrix defined as a 4x4 matrix for 4 samples, but the example makes no mention of whether these are empirical or augmented samples.

Discussion in the appendix indicates that the work follows that of HaoChen et al. (2021), where the adjacency matrix is defined as NxN for N total augmented samples. Does this model of difficult examples make no distinction between samples of the same class and samples augmented from the same empirical sample?

Typo in section 5.1? According to equation 12, posHigh is supposed to be greater than posLow, but Figures 4(a) and 4(b) denote otherwise.

Plot the bounds in Theorems 3.1 and 3.2 for values of $\beta, \gamma$, and $\alpha$ for a clear comparison.

Unless I am mistaken, some old and some recent papers show that hard-negative sampling improves performance in contrastive learning - do the theoretical claims in the paper challenge this observation?

It will be good to see the results on purely synthetic data - say a mixture of Gaussians where each mixture is a class. Can the authors produce a table for this case?

---

> ### Author Response · Authors · 2025-11-21
>
> We thank Reviewer Vvxq for the insightful and positive comments. We appreciate the recognition of the rigor of our mathematical framework and its implications for sample selection in contrastive learning. We address each of your comments in detail below.
>
> ---
>
> **Q1.** The experimental section is only partially sound, with a limited number of experiments and hyperparameter tuning.
>
> **A1.** We would like to clarify that the four hyperparameters — $\sigma$, $\rho$, and $Sim_{posHigh}$ / $Sim_{posLow}$ — are intrinsic components of our method rather than parameters inherited from prior work. **We have performed ablation studies on each of them**, and the corresponding results are reported in Figures 4(a,b) and 5(a,b). These results show that a broad range of values consistently leads to improvements, indicating that our method is not sensitive to hyperparameter tuning.
>
> Beyond the standard experimental setup, **we additionally evaluated our approach across multiple self-supervised learning frameworks, diverse datasets of varying difficulty, and long-tailed distributions.** In all cases, the proposed method improves the baseline, demonstrating robustness and general applicability. We hope these additional results fully address the reviewer’s concerns about the experimental soundness.
>
>
> ---
>
> **Q2.**  The presentation needs to be significantly improved. There are typographical errors throughout the paper. Figures 1, 2, 4 contain text that is too small to read. Certain ideas are not explained clearly enough.
>
>
> **A2.** We sincerely appreciate your valuable comments. In the revised version, we have carefully addressed this. Specifically, **we have enlarged the fonts in Figures 1, 2, and 4** to improve readability, and we hope this makes the figures clearer and easier to follow.
>
> ---
>
> **Q3.** The theoretical framework closely follows HaoChen et al. (2021), which already discusses similarities and differences between empirical and augmented sets. How does the current paper provide additional insights or improvements beyond this existing work?
>
> **A3.** Although closely following HaoChen et al. (2021), our work focuses on the effects of difficult-to-learn examples, which has not been previously studied. Specifically, our framework is based on a specific similarity graph that characterizes difficult-to-learn samples, **with the goal of explaining their impacts (Section 3.3) and proposing possible solutions (Section 4).** In contrast, Haochen’s work relies on a general augmentation graph aimed at deriving a downstream error bound.
>
> ---
>
> **Q4.** The results of experimentation in Section 5 are only averaged out of 3 runs. This makes most results (almost all that are not on TinyImagenet) close enough to be within margin of error. This is exacerbated by the fact that, as shown in Appendix A.3-A.4, only limited hyperparameter tuning is done. Moreover, TinyImagenet is the only dataset that uses a ResNet-50 model instead of ResNet-18 - a natural question arises: could the performance increase be related to the complexity of the model?
>
> **A4.** To better demonstrate the effectiveness of our method, we report the complete results over three runs on CIFAR-100 as an example.
>
> | **Run** | **SimCLR** | **MT** | **TS** | **Combined** |
> |----------|------------------------|---------------------------|---------------------------|------------------------|
> | **Run 1** | 59.81 | 61.11 | 61.49 | 62.69 |
> | **Run 2** | 60.05 | 61.33 | 61.82 | 63.02 |
> | **Run 3** | 60.00 | 61.40 | 61.70 | 62.87 |
> | **Mean**  | **59.95** | **61.28** | **61.67** | **62.86** |
> | **STD**  | **0.13** | **0.15** | **0.17** | **0.17** |
>
> From the results, we can observe that our method consistently achieves performance improvements over the baseline. For the discussion on hyperparameter tuning, please refer to our response to Q1.
>
> Regarding the results on Tiny-ImageNet, we agree with your understanding. As Tiny-ImageNet is a more challenging dataset compared to smaller ones such as CIFAR-10/100, we used the largest backbone feasible within our computational budget, ResNet-50, while ResNet-18 was used for the simpler datasets. Indeed, since difficult samples inevitably exist in real-world datasets, models with stronger representation capacity are naturally more compatible with our method, as they can characterize inter-sample similarities more effectively.

---

> ### Author Response · Authors · 2025-11-21
>
> **Q5.** What exactly does the mixing process look like in the mixing image experiment (Section 2)?  Please provide a clearer description or example.
>
> **A5.** We clarify that the γ-Mixed CIFAR-10 dataset refers to a modified version of CIFAR-10, in which a proportion γ of the samples are mixed at the pixel level. For instance, an image of a cat and an image of a dog are blended together to create a new sample that lies closer to the decision boundary between these two classes. **This process effectively simulates intrinsically difficult samples near the class boundary.** To better illustrate this concept, we provide an anonymized visualization: in the image here https://imgur.com/a/xRabrzv, the right example shows the mixed image generated by our procedure.
>
> ---
>
> **Q6.** Section 3.2 defines the adjacency matrix as a $4 \times 4$ matrix for 4 samples, but it is unclear whether these are empirical or augmented samples.
>
> **A6.** They are augmented samples, just as in the augmentation graph of HaoChen et al. (2021). For more details, we present the formal definition of an $n\times n$ adjacency matrix in Appendix B. The $4 \times 4$ matrix in Section 3.2 is just for better illustration of our mathematical modeling.
>
> ---
>
> **Q7.** Discussion in the appendix indicates that the work follows that of HaoChen et al. (2021), where the adjacency matrix is defined as NxN for N total augmented samples. Does this model of difficult examples make no distinction between samples of the same class and samples augmented from the same empirical sample?
>
> **A7.** We did not make such a distinction, because the theoretical analysis aims to study the role of difficult-to-learn examples, focusing mainly on the similarity between samples from different classes. Therefore, we did not differentiate between samples of the same class or augmented versions of the same sample, as this is not the main focus of our work.
>
> ---
>
> **Q8.** Typo in section 5.1? According to equation 12, posHigh is supposed to be greater than posLow, but Figures 4(a) and 4(b) denote otherwise.
>
> **A8.** The parameters posLow and posHigh correspond to specific percentiles of cosine similarity values, ranked from high to low. For example, setting posLow = 0.3 means selecting the top 30% of pairs with the highest cosine similarity, while posHigh = 0.1 corresponds to the top 10%.
> Accordingly, **the similarity values satisfy the relationship $Sim_{posLow} < Sim_{posHigh}$, which naturally follows from their percentile definitions.**
>
> ---
> **Q9.** Plot the bounds in Theorems 3.1 and 3.2 for values of $\beta$, $\gamma$, and $\alpha$ for a clear comparison.
>
> **A9.** Following your suggestion, we plot the value difference of the two bounds w.r.t. different $\alpha$, $\beta$, and $\gamma$ values to illustrate the comparison between the two bounds. We provide an anonymized visualization: https://imgur.com/a/Qe6fKvX. In the image, for given values of $\alpha=0.6$ (upper) and $0.8$ (lower), we plot the value difference between the two bounds $\text{Bound}\_{\text{w.d.}} - \text{Bound}\_{\text{w.o.}}$ (right) and the sign of the difference $\text{sgn}(\text{Bound}\_{\text{w.d.}} - \text{Bound}\_{\text{w.o.}})$ (left), w.r.t. different values of $\beta \in (0,\alpha)$ and $\gamma \in (0,\alpha)$. To calculate the bound values, we take the sample size $n=50000$, number of classes $(r+1)=10$, and $n_d=8\% \times n$ (where the settings are compatible with the CIFAR-10 dataset). From the left subfigures, we show that when $\gamma>\beta$ (the upper triangle part of each subfigure), the error bound with difficult examples is larger than the bound without difficult examples, and from the right subfigures, we can see that larger $\gamma-\beta$ result in greater differences of bound values. Also, by comparing the upper and lower subfigures, we see that different $\alpha$ values do not change this conclusion. In other words, **this visualization shows that when $\alpha>\gamma>\beta$, we have $\text{Bound}\_{\text{w.d.}} > \text{Bound}\_{\text{w.o.}}$, which coincides with the insight in our paper.**

---

> ### Author Response · Authors · 2025-11-21
>
> **Q10.** Unless I am mistaken, some old and some recent papers show that hard-negative sampling improves performance in contrastive learning - do the theoretical claims in the paper challenge this observation?
>
> **A10.** In fact, **our study focuses on a different concept from hard negatives.** While both difficult examples and hard negative samples influence the performance of self-supervised learning, they are defined from different perspectives: difficult examples relate to the classification boundary, whereas hard negatives are defined with respect to the anchor point in contrastive learning. Therefore, **the two concepts are fundamentally different and do not challenge or contradict each other.**
>
> ---
>
> **Q11.** It will be good to see the results on purely synthetic data - say a mixture of Gaussians where each mixture is a class. Can the authors produce a table for this case?
>
> **A11.** We appreciate your suggestion of evaluating on purely synthetic data such as Gaussian mixtures. However, representations extracted from Gaussian-distributed points lack the representative and semantically meaningful structures required for contrastive learning, making this setting not an informative evaluation for our method. **To better reflect the intention of assessing robustness under synthetic-like scenarios, we instead constructed the Mixed CIFAR dataset, which simulates mixed samples while preserving meaningful semantics and augmentability. As shown in Figure 6, our method consistently outperforms the baseline across all mixing ratios, validating its effectiveness in synthetic-like settings.**
>
> ---
>
> We thank the reviewer once again for the thoughtful feedback and constructive suggestions, which we believe have helped make the paper clearer and stronger.
>
> ---

---

### Official Review · Reviewer_Dph8 · 2025-10-31

**Soundness:** 3
**Presentation:** 3
**Contribution:** 2
**Rating:** 6
**Confidence:** 3

**Summary:**

The paper analyzes when and why "difficult" examples can hurt contrastive representation learning. It models pairwise similarities with a block structure using within-class, easy cross-class, and difficult cross-class.
Within a spectral view of CL, the authors derive linear probe error bounds showing that adding difficult examples worsens the bound. Conversely, removing them or correcting pairwise affinities via pair-specific margin tuning or temperature scaling improves it.
Empirically, targeted adjustments on the selected pairs yield consistent gains across CIFAR-10/100, STL-10, and Tiny-ImageNet.

**Strengths:**

1. The paper theoretically explains the counterintuitive phenomenon that adding "hard" examples can hurt contrastive learning. The spectral framework is a natural fit and the analysis is clean. The block model exposes the role of $\gamma-\beta$ on linear-probe error.
1. The authors propose practical interventions of margin and temperature adjustments motivated by the theory, with improved theoretical error bound. Consistent empirical improvements are observed when applying the interventions. Furthermore, combining selection with targeted adjustments produces the largest gains across datasets.

**Weaknesses:**

1. The theory assumes $0 \leq \beta < \gamma < \alpha < 1$, yet cosine similarity lies in $[-1,1]$. Are similarities shifted or computed from a PSD kernel where values are guaranteed to be nonnegative? If not, can the theorems be relaxed to allow $\beta,\gamma<0$?
1. A brief discussion of the tightness of the presented bounds would be helpful in understanding the significance of the bound. Context on any known lower bounds or contrasting constructions, or even a heuristic level argument would help.
1. Since the improvements are modest except for Tiny-ImageNet, it would be helpful to see the variation across multiple runs. I am also curious if the proposed adjustments can work for other modalities such as text.

**Minor comments**
1. Please provide a precise recipe for the $\gamma$-Mixed CIFAR-10 dataset.
1. Assumptions should appear prominently in the main body with a brief discussion of plausibility, as they are central to the theoretical results.
1. Section 5.1 needs a clearer explanation.
1. The writing can be improved. Some examples are as follows:
  - In Section 4.2 (Margin Tuning), the method can be briefly introduced with a math expression.
  - Similarly, in Section 4.3, temperature scaling can be briefly introduced with a math expression.
  - In Corollary 4.1, should the $\alpha$ appearing in the numerator be $\delta$?
  - Figure 3 is not helpful in understanding the role of $\alpha$, $\beta$, and $\gamma$.
  - In the definition of $w_{x,x'}$ in line 142, $\mathbb{E}\tilde x \sim \bar{\mathcal{P}}$ should be $\mathbb{E}_{\tilde x \sim \bar{\mathcal{P}}}$, where $\bar{\mathcal{P}}$ seems undefined.
  - The spectral contrastive loss defined in Eq. (1) has an argument $\mathbf{x}$, which seems unnecessary.
  - In line 288, there is a missing space between "pairs." and "Here".
  - In line 101: possibly "preciously" is a typo for "precisely".
  - In lines 1530 and 1540, theorem numbers are missing in the titles of each theorem. Also a typo: "erorr" should be "error".
  - Theorem B.7 defines $\mathcal{E}{w.d.}$, whereas the inequality has $\mathcal{E}{w.o.}$.
  - Line 1537: "folowing" $\rightarrow$ "following".
  - Line 163: "decision difficult" $\rightarrow$ "decision boundary".
  - Line 331: "an combined method" $\rightarrow$ "a combined method".
  - Title of Section 5.4: "DIFFICLUT" $\rightarrow$ "DIFFICULT".
  - Lines 743 and 748: "Eq. equation 12" $\rightarrow$ "Eq. 12".

**Questions:**

1. Can the theory be relaxed to allow negative $\beta$ and $\gamma$?
1. Is it possible to show the lower bound for the $\mathcal{E}_{w.d}$?
1. Tiny-ImageNet shows much larger gains under the targeted edits than CIFAR-10/100 and STL-10 (Table 2 and 3). I am curious on what derives this difference.

---

> ### Author Response · Authors · 2025-11-21
>
> We thank Reviewer Dph8 for the encouraging and insightful comments. We appreciate the recognition of our theoretical analysis and the effectiveness of our proposed interventions. We address each of your comments in detail below.
>
> ---
>
> **Q1.** The theory assumes $0 \le \beta \lt \gamma \lt \alpha \lt 1$, yet cosine similarity lies in $[-1, 1]$.  Are similarities shifted or computed from a PSD kernel where values are guaranteed non-negative?  If not, can the theorems be relaxed to allow $\beta, \gamma < 0$?
>
> **A1.** The normalization guarantees the largest similarity value to be $1$, but cannot guarantee non-negativity. **In fact, the theorems do allow $\beta, \gamma < 0$ as long as $\alpha \geq \gamma \geq \beta$.** We have revised our manuscript and removed the non-negative condition.
>
> ---
>
> **Q2.**  A brief discussion of the tightness of the presented bounds would be helpful in understanding their significance.  Context on any known lower bounds, contrasting constructions, or heuristic arguments would also be useful. Is it possible to show the lower bound for the $\mathcal{E}_{w,d}$?
>
>
> **A2.** As a possible extension, we discuss that a potential lower bound related to the $\lambda_{k}$, which according to similar proofs of Theorem 3.2, also increases with $\gamma-\beta$ increasing. Besides, in the appendix, **we empirically verify that the trending of the downstream classification error, which also verifies the reliability of our upper bounds.** Please see the following for more details.
>
> To derive the potential lower bound, we are mainly required to modify Theorem B.6 in HaoChen et al. (2021) to an alternative "lower bound" form (because our analysis is based on Theorem B.3 in HaoChen et al. (2021) and a key step to its proof is Theorem B.6). Specifically, with the notations of Theorem B.6 (while replacing $\lambda_i$ with $1-\lambda_i$ to be consistent with our submission), we have
> $$
> \\|u - \sum\_{i=1}^k b_iv_i\\|\_2^2
> = \sum\_{i=k+1}^N \zeta_i^2
> \geq \frac{1}{(1-\lambda\_{N})} [\sum\_{i=1}^N (1-\lambda_i) \zeta_i^2 - \sum\_{i=1}^k (1-\lambda_i) \zeta_i^2]
> \geq \frac{1}{(1-\lambda\_{N})} [R\|u\|\_2^2 - (1-\lambda_k) \sum\_{i=1}^k\|b_i\|^2].
> $$
> According to Fulton (2000), we have $\lambda_k \geq \max_{i+j=n+k} \lambda_{1,i} + \lambda_{2,j}$, that is, we have $\lambda_N \geq [(1-\alpha)-r(\gamma-\beta)]/c_1$ and $\lambda_k \geq [(1-\alpha)+r(\gamma-\beta)]/c_1$ for $k\leq n_d$. Additionally, by Claim B.7 and Lemma B.8 in HaoChen et al. (2021) and some additional assumption that $\sum_{i=1}^k\|b_i\|^2\leq C$, where $C>0$ is a constant, we obtain the lower bound as
> $$
> \mathcal{E}_{\mathrm{w.d.}} \geq \frac{1}{1-\frac{(1-\alpha)-r(\gamma-\beta)}{c_1}}\Big[\frac{1}{2}\phi^{\hat{y}} - 1 + \frac{(1-\alpha)+r(\gamma-\beta)}{c_1}C\Big].
> $$
> Note that $\frac{1}{1-\frac{(1-\alpha)-r(\gamma-\beta)}{c_1}}$ decreases and $\frac{1}{1-\frac{(1-\alpha)-r(\gamma-\beta)}{c_1}} \cdot \frac{(1-\alpha)+r(\gamma-\beta)}{c_1}$ increases as $\gamma-\beta$ increases. Then when the labeling error $\phi^{\hat{y}}$ is relatively small, the lower bound gets larger with $\gamma-\beta$ increasing, which coincides with the insight of Theorem 3.2 in our submission.
>
> In addition, in Table 10, we empirically show the trending of our derived bounds, which also to some extent indicates the tightness of bounds. Specifically, **on a Mixed CIFAR synthetic dataset, we show that as the difficult examples increase, the classification error and the bound share the same variation trend, thus validating Theorem 3.2 that a larger $\gamma-\beta$ results in worse error.**

---

> ### Author Response · Authors · 2025-11-21
>
> **Q3.** The improvements are modest except for Tiny-ImageNet. It would be helpful to show results over multiple runs, and to evaluate whether the proposed adjustments can generalize to other modalities such as text.
>
> **A3.** To better demonstrate the effectiveness of our method, we report the complete results over three runs on CIFAR-100 as an example.
>
> | **Run** | **SimCLR** | **MT** | **TS** | **Combined** |
> |----------|------------------------|---------------------------|---------------------------|------------------------|
> | **Run 1** | 59.81 | 61.11 | 61.49 | 62.69 |
> | **Run 2** | 60.05 | 61.33 | 61.82 | 63.02 |
> | **Run 3** | 60.00 | 61.40 | 61.70 | 62.87 |
> | **Mean**  | **59.95** | **61.28** | **61.67** | **62.86** |
> | **STD**  | **0.13** | **0.15** | **0.17** | **0.17** |
>
> From the results, **we can observe that our method consistently achieves performance improvements over the baseline.**
>
> We appreciate your valuable suggestion. In this part, we adopted SimCSE, a widely used self-supervised contrastive learning algorithm in the text modality. We used the Yahoo Answers Topics dataset (a text classification corpus) as our unsupervised training data. Specifically, we sampled 10,000 instances from each category as the training set and used bert-base-uncased as the underlying encoder model.
>
> We evaluated four configurations:
> (1) the original SimCSE baseline,
> (2) SimCSE with margin tuning,
> (3) SimCSE with temperature scaling, and
> (4) the combination of both methods.
>
> Concretely, we set poslow = 0.25, poshigh = 0.13, and both σ and ρ to 0.1, and trained each model until convergence. We then conducted Linear Probe classification on Yahoo Answers Topics to assess the quality of the learned sentence embeddings.
>
>
> | Method  | **SimCSE** | **MT** | **TS** | **Combined** |
> |----------|------------------------|---------------------------|---------------------------|------------------------|
> | **Linear Probe Acc**  | **68.05** | **69.57** | **69.84** | **70.22** |
>
> As shown in the table, **our method achieves consistent performance improvements in the text modality.**
>
> ---
>
> **Q4.** Please provide a precise recipe for constructing the γ-Mixed CIFAR-10 dataset.
>
> **A4.** We clarify that the γ-Mixed CIFAR-10 dataset refers to a modified version of CIFAR-10, in which a proportion γ of the samples are mixed at the pixel level. **For instance, an image of a cat and an image of a dog are blended together to create a new sample that lies closer to the decision boundary between these two classes.** This process effectively simulates intrinsically difficult samples near the class boundary. To better illustrate this concept, we provide an anonymized visualization: in the image here https://imgur.com/a/xRabrzv, the right example shows the mixed image generated by our procedure.
>
> ---
>
> **Q5.** Assumptions should appear more prominently in the main body, accompanied by a brief discussion of their plausibility,
> as they are central to the theoretical results.
>
> **A5.** Following your suggestions, in our revised version, we place the formal Assumptions and brief discussions at the beginning of Section 3.3. They were initially placed in the appendix due to page limit.
>
>
>
> ---
>
> **Q6.** Section 5.1 needs a clearer explanation.
>
> **A6.**
> Section 5.1 presents a principled mechanism for identifying difficult sample pairs. Instead of relying on additional models or auxiliary supervision, we define difficult pairs as samples that come from different classes while remaining highly similar in the representation space, and we introduce an efficient percentile-based similarity selection mechanism to identify them. Figure 4 further confirms that this selection mechanism remains stable across training.
>
> ---
>
> **Q7.**
> The writing can be improved.
>
> **A7.**
> We sincerely appreciate your valuable comments, and we have carefully corrected them in the revised version.
>
> ---
>
>
>
>
> **Q8.** Tiny-ImageNet shows much larger gains than CIFAR-10/100 and STL-10 (Tables 2–3).
> What drives this discrepancy?
>
>
> **A8.** Tiny-ImageNet is substantially more challenging than CIFAR-10/100 and STL-10 due to its larger number of categories and higher intra-class variation. To accommodate its complexity, we used the largest backbone feasible within our computational budget (ResNet-50), whereas ResNet-18 was used for the smaller datasets. This architectural difference largely explains the larger performance gain on Tiny-ImageNet, because a stronger backbone provides more representational capacity for our method to leverage. Importantly, across all datasets and backbone configurations, our method consistently improves the baseline, indicating that the effectiveness of the proposed approach is not tied to a specific dataset or architecture.
>
> ---
>
>
> We thank the reviewer once again for the thoughtful feedback and constructive suggestions, which we believe have helped make the paper clearer and stronger.

---

### Official Review · Reviewer_3ZD8 · 2025-10-31

**Soundness:** 3
**Presentation:** 2
**Contribution:** 3
**Rating:** 6
**Confidence:** 5

**Summary:**

Paper proves difficult examples (near decision boundaries) harm unsupervised contrastive learning, contrary to supervised learning. Develops similarity graph framework with α, β, γ parameters modeling sample pair relationships. Derives generalization bounds showing difficult examples worsen performance. Proposes three mitigation strategies—sample removal, margin tuning, temperature scaling—validated theoretically and empirically on CIFAR-10/100, STL-10, TinyImagenet.

**Strengths:**

- **Novel counterintuitive finding**: Removing 8-20% of training data improves performance consistently across 4 datasets

- **Rigorous theoretical framework**: Clean similarity graph model (Theorems 3.1-3.2) proves difficult examples increase error bound from 4δ/(1-λ) to 4δ/(1-λ') where λ' > λ

- **Provided solutions with theory**: Margin tuning (Theorem 4.3), temperature scaling (Theorem 4.5), removal (Corollary 4.1) all improve bounds; experiments confirm (Table 4: Combined method +1.6% CIFAR-10, +4.9% CIFAR-100)

- **Comprehensive extensions**: MoCo (Table 5), long-tail (Table 6), relaxed assumptions (Appendix B.3), ImageNet-1K (Table 8)

**Weaknesses:**

- **Circular difficult example definition**: Section 5.1 selects "currently confusing" examples via cosine similarity during training, not "intrinsically difficult" ones. Figure 4c shows ratio evolves to 90%+ suggesting selection is training-dependent

- **Scalability concerns**: ImageNet-1K gains only 1.36% (Table 8) vs 2-15% on smaller datasets. Only 400 epochs vs standard 800+. No computational cost analysis in Algorithm 1

- **Hyperparameter theory disconnect**: Theorems 4.3/4.5 derive exact margins (Eq. 7) and temperatures (Eq. 10) but experiments use simplified σ, ρ without justification. Dataset-dependent tuning required (Appendix A.3)

- **Theorem 3.2 condition unclear**: Requires nd ≤ k ≤ nd + r + 1 but k (feature dim) is architecture choice, not dataset property. Restrictiveness in practice unexplained

**Questions:**

1. **Hyperparameters**: Why use simplified σ, ρ instead of theoretically optimal values from Eq. 7, 10? What performance is lost?

2. **Hard negatives**: How do your methods compare against Robinson et al. 2020 and Kalantidis et al. 2020? Appendix A.1 claims distinction but no empirical comparison

3. **Scalability**: Why do ImageNet gains (1.36%) diminish vs CIFAR (2-15%)? Is this fundamental or due to 400 vs 800 epochs? What is computational overhead?

4. **Selection stability**: Figure 4c shows selection evolves during training. Are you removing "intrinsically difficult" or just "not-yet-learned" examples? Would different selection mechanisms yield same results?

5. **Theorem 3.2 condition**: When does nd ≤ k ≤ nd + r + 1 restrict practical applicability? What happens outside this range?

---

> ### Author Response · Authors · 2025-11-21
>
> We sincerely thank Reviewer 3ZD8 for the positive and encouraging feedback. We appreciate the recognition of our work’s novelty, theoretical rigor, and consistent empirical validation across multiple datasets and settings. We address each of your comments in detail below.
>
> ---
> **Q1. Circular Difficult Example Definition.** Section 5.1 selects currently confusing examples via cosine similarity during training, not “intrinsically difficult.” Figure 4c shows ratio evolves to 90%+, suggesting selection is training-dependent.
>
> **A1.** The definition of difficult examples is theoretical rather than training-dependent. **Difficult examples refer to samples close to the Bayesian decision boundary, meaning they are intrinsically hard to distinguish.** Figure 4c does not imply that difficulty is induced by training; instead, it verifies that the selected region indeed corresponds to intrinsically difficult pairs. Specifically, even at the very beginning of training, about 89% of the selected pairs come from different classes while exhibiting high similarity in the representation space, which is fully consistent with the definition of intrinsically difficult examples.
>
> ---
> **Q2. Scalability Concerns.** ImageNet-1K gains only 1.36% (Table 8) vs ≥2–15% on smaller datasets. Only 400 epochs vs standard 800+. No computational cost analysis in Algorithm 1.
>
> **A2.** Due to limited computational resources, we downscaled the ImageNet-1K images to 96×96, which inevitably reduced image resolution and made feature separation more challenging. Additionally, even after compression, the 96×96 input size remains significantly larger than those used in TinyImageNet and CIFAR datasets. Constrained by memory limitations, we could only employ ResNet-18 as the backbone network for this large-scale dataset. The combination of a relatively shallow network and lower-resolution inputs resulted in smaller performance gains compared to those achieved on smaller datasets. Nevertheless, even under these challenging conditions with only 400 training epochs, our method still achieves a 1.36% performance improvement. We believe that with sufficient computational resources—enabling the use of higher-resolution inputs and deeper network architectures—our approach would achieve more substantial gains on large-scale datasets as well.
>
> To address the computational-cost concern, we compared the runtime of our method with SimCLR under the same training configuration. On CIFAR-100, the training time for 100 epochs is:
>
> | **Methods** | **Baseline** | **Ours** |
> |--------------|--------------|-----------|
> | **Seconds**  | 2035     | 2046   |
>
> Our method introduces only +0.54% overhead, indicating that **the proposed mechanism maintains the computational efficiency of the baseline.**
>
> ---
> **Q3. Hyperparameter–Theory Disconnect.** Theorems 3.4/4.5 derive exact margins (Eq. 7) and temperatures (Eq. 10), but experiments use simplified σ, ρ without justification.  Dataset-dependent tuning required (Appendix A.3).
>
> **A3.** Theoretical parameters $\alpha$, $\beta$ and $\gamma$ describe the ideal similarity between augmented samples. **These theoretical quantities are not directly observable in real dataset, and their estimations may require introducing additional pretrained encoders.** Therefore, we use tunable parameters $\sigma$ and $\rho$ as empirical proxies for the theoretically derived quantities in Equations (7) and (10). In this sense, $\sigma$ and $\rho$ are not arbitrary replacements but an operational instantiation of the theoretical margins derived in Theorems 4.3 and 4.5. Moreover, Figures 5a and 5b show that a wide range of $\sigma$ and $\rho$ values yield consistent performance improvements, indicating that the method does not rely on specific tuning.
>
> ---
> **Q4. Theorem 3.2 condition.** When does nd ≤ k ≤ nd + r + 1 restrict practical applicability? What happens outside this range?
>
> **A4.** We first apologize for the typo here. The range should be $r+1 \leq k < n_d+r+1$ as evidenced by the proof. In practice, the number of classes are considered as a fixed number and typically the embedding dimension $k$ is larger than the number of classes. On the other hand, the sample size is usually large enough, so the number of negative samples $n_d$ as a certain fraction of $n$ is also large enough so that the embedding dimension $k < n_d+r+1$. For example, in experiments, we select $8\%$ hard negative samples, so $n_d = 8\%\times 50000=4000$ even for the smallest CIFAR-10 dataset, which is significantly larger than the embedding dimension $k=128$ (after projection head).
>
> Nonetheless, according to the proofs regarding the eigenvalues, we also note that when $k \leq r$, the bound becomes
> $$
> \mathcal{E} \leq \frac{4\delta}{1 - (\frac{1}{c_1}(1-\alpha) + \frac{r}{c_1}(\gamma-\beta) + (\alpha-\beta)\Big[\frac{\kappa-1}{c_2} + \frac{1}{c_1}\Big]n_d)}+8\delta,
> $$
> which also becomes worse as the difficult examples become more challenging (larger $\gamma-\beta$).

---

> ### Author Response · Authors · 2025-11-21
>
> ---
>
> **Q5. Hard Negatives.** How do your methods compare against Robinson et al. (2020) and Kalantidis et al. (2020)?  Appendix A.1 claims distinction but no empirical comparison is provided.
>
>
> **A5.** According to your suggestion, we reproduced the representative hard-negative-based method HCL proposed by Robinson et al. (2020) and evaluated it under the same experimental settings on CIFAR-100. The experimental results are shown below:
>
> | **Method** | **SimCLR** |**HCL** | **Ours** |
> |-------------|----------|-----------|----------|
> | **Performance** |  59.95 | 62.13 | 62.86 |
>
> The results show that although difficult examples and hard negative samples are defined from different perspectives, where difficult examples relate to the classification boundary and hard negatives are defined with respect to the anchor point in contrastive learning, **both paradigms are beneficial to contrastive learning**.
>
>
>
> ---
>
> **Q6. Selection Stability.** Figure 4c shows selection evolves during training.  Are you removing “intrinsically difficult” or just “not-yet-learned” examples? Would different selection mechanisms yield same results?
>
> **A6.** Our selection mechanism targets intrinsically difficult sample pairs rather than merely not-yet-learned ones. As shown in Figure 4c, the selected ratio is already high at the beginning of training, **demonstrating that the mechanism can identify intrinsically difficult pairs without needing a warm-up stage.**  Although the ratio increases slightly as training progresses, the strong initial accuracy indicates that the selection is not driven by transient “not-yet-learned” noise but reflects inherent difficulty.
>
> Regarding whether different selection mechanisms would yield the same results, our method is specifically designed to select intrinsically difficult sample pairs. While selecting “not-yet-learned” examples may also be of interest for future work, it represents a different objective from the focus of this paper, which is to identify samples that are inherently difficult rather than temporarily unlearned.
>
> ---
>
> We thank the reviewer once again for the thoughtful feedback and constructive suggestions, which we believe have helped make the paper clearer and stronger.

---

### Author Response · Authors · 2025-11-29
**Rebuttal Summary and Authors' Response**

**Dear Program Chairs, Senior Area Chairs, Area Chairs, and Reviewers,**

We sincerely appreciate the tremendous efforts of the **Program Chairs, Senior Area Chairs, and Area Chairs** in coordinating the review process. We also extend our sincere thanks to all **Reviewers** for their constructive and detailed feedback.

We are greatly encouraged that the reviewers have recognized the value of our work, specifically highlighting the "**novel counterintuitive finding**" that difficult examples can harm contrastive learning (Reviewers 3ZD8 & Dph8), the "**rigorous**" and "**sound**" theoretical framework where the "**analysis is clean**" (all Reviewers), and the "**consistent empirical improvements**" achieved by our proposed interventions (Reviewers 3ZD8 & Dph8).

During the rebuttal phase, we have provided detailed point-by-point responses to all comments raised by the reviewers. **Here, we provide a concise overview of the key improvements below:**

**1. Experimental Enhancements**
To demonstrate the robustness, generality, and efficiency of our method, we have added significant new results:
* **Generalization to Text Modality (Response to Reviewer Dph8):** We extended our evaluation to the NLP domain using SimCSE on the Yahoo Answers dataset. Our method consistently improved the baseline (**+2.17%**), validating its applicability beyond computer vision.
* **Comparison with Hard Negatives (Response to Reviewer 3ZD8):** We conducted a direct comparison with HCL (Robinson et al., 2020) on CIFAR-100. Results show that both methods improve upon the SimCLR baseline. This demonstrates that although "difficult examples" and "hard negatives" are defined from different perspectives, **both paradigms are beneficial for contrastive learning.**
* **Statistical Stability (Response to Reviewers Vvxq & Dph8):** We reported results over **multiple independent runs** on CIFAR-100. The low standard deviation (e.g., **0.17**) confirms the statistical significance and reproducibility of our gains.
* **Construction of γ-Mixed Dataset (Response to Reviewers Dph8 & Vvxq):** We provided a precise recipe and visualization for constructing the γ-Mixed CIFAR-10 dataset, explicitly showing how image mixing effectively simulates intrinsically difficult samples near the decision boundary.
* **Computational Efficiency (Response to Reviewer 3ZD8):** We empirically verified the efficiency. On CIFAR-100, our method introduces a **negligible overhead** of only **+0.54%** training time compared to the baseline, proving excellent scalability.

**2. Theoretical Clarifications**
We have refined our theoretical analysis to ensure rigor and clarity:
* **Condition Clarification (Response to Reviewer 3ZD8):** We clarify the condition of Theorem 3.2, discussing how this condition holds in practice and the potential result beyond this condition.
* **Relaxation of Assumptions (Response to Reviewer Dph8):** We clarified that our theorems hold even when cosine similarity values are negative, and we have relaxed the non-negativity condition in the revised manuscript.
* **Bound Tightness (Response to Reviewer Dph8):** To verify the bound tightness, we theoretically discuss a potential lower bound that has the same trending as the upper bound in our submission, and we also clarify the related empirical verifications.
* **Distinction from Prior Art (Response to Reviewer Vvxq):** We clarified the theoretical boundary between our "difficult-to-learn" framework and the general augmentation graph in HaoChen et al. (2021), emphasizing our specific contribution to analyzing difficult samples.
* **Visualization for bound comparison (Response to Reviewer Vvxq):** We plot the bound differences under varying α, β, and γ values for a clear comparison.

**3. Presentation Improvements**
We have comprehensively revised the manuscript to enhance clarity and presentation quality:
* **Visual Readability:** As suggested by Reviewer Vvxq, we have **enlarged the font sizes** in Figures 1, 2, and 4 to ensure optimal legibility.
* **Corrections:** We carefully corrected typographical errors pointed out by the reviewers.

**Beyond these major updates, we have also addressed other detailed inquiries in our individual responses, such as clarifying that our selection mechanism identifies intrinsically difficult pairs, further explaining the consistent effectiveness across diverse datasets, and verifying the robustness to hyperparameter tuning.**

We believe these revisions have significantly strengthened the paper. **We respectfully hope that this summary is helpful.**

Thank you once again for your dedication to the community.

Best regards,

**Authors**

---

### Meta-Review · Area_Chair_pL4S · 2026-01-07

**Summary:**

I found this paper to be very interesting, particularly in how it highlights the divergent behaviors of unsupervised learning compared to supervised learning. The reviewers have identified significant strengths in the work. For instance, Reviewer 3ZD8 highlighted the novelty of the counterintuitive finding that difficult examples can hurt generalization. The theoretical framework was praised as beautiful and sound by Reviewers 3ZD8, Dph8, and Vvxq, and Reviewers 3ZD8 and Dph8 specifically noted that the proposed solution is well-grounded in this theory. Furthermore, Reviewer 3ZD8 commended the extensive verification provided by the authors.

However, I would like to draw the authors' attention to a critical nuance regarding the definition of "difficult examples." The current interpretation appears to lean heavily on definitions derived from supervised learning. It is worth noting that the nature of difficulty in self-supervised learning may be fundamentally different. If we were to redefine "difficult examples" to correctly align with self-supervised paradigms, it is possible that the results would align more closely with the beneficial trends seen in supervised learning. For example, there is existing literature suggesting that even a single image can be sufficient for self-supervised learning. Such instances might represent the true "difficult examples" in this specific domain, and this kind of difficulty could potentially help rather than hinder the learning process. Additionally, the excluding observation on the STL-10 dataset needs a more thorough explanation.

The reviewers also raised some valid concerns that should be addressed. Reviewers 3ZD8 and Dph8 requested clarifications regarding definitions, scalability, hyperparameters, and variance. Reviewer Dph8 also asked for theoretical clarifications regarding non-negative aspects and bound tightness, while Reviewer 3ZD8 noted the reliance on existing mathematical tools. Addressing these points, alongside the core issue of defining difficulty in the context of SSL, would significantly strengthen the final version of the paper.

**Reviewer Concerns:**

I believe the authors have addressed most of the concerns in a fair and adequate manner.

**Reviewer Scores:**

I expect that all reviewers will maintain an accept recommendation for this paper. Some may choose to increase their scores, while others may keep their current evaluations; at the very least, a change from accept to reject seems unlikely. This assessment assumes that the review process remains fair and consistent.

---

### Decision · Program_Chairs · 2026-01-26

Accept (Oral)